# Estimation of Electrical Conductivity and Magnetization Parameter of Neutron Star Crusts and Applied to the High-Braking-Index Pulsar PSR J1640-4631

**Hui Wang [1,2,3], Zhi-Fu Gao [1,2,*], Huan-Yu Jia [3], Na Wang [1] and Xiang-Dong Li [4,5]**

[1] Xinjiang Astronomical Observatory, Chinese Academy of Sciences, 150, Science 1-Street, Urumqi 830011, China; gjwang@my.swjtu.edu.cn (H.W.); na.wang@xao.ac.cn (N.W.)

[2] Key Laboratory of Radio Astronomy, Chinese Academy of Sciences, West Beijing Road, Nanjing 210008, China

[3] School of Physical Science and Technology, Southwest Jiao tong University, Chengdu 610031, China; hyjia@swjtu.cn

[4] School of Astronomy and Space Science, Nanjing University, Nanjing 210023, China; lixd@nju.edu.cn

[5] Key Laboratory of Modern Astronomy and Astrophysics (Nanjing University), Ministry of Education, Nanjing 210023, China

* Correspondence: zhifugao@xao.ac.cn

**Abstract:** Young pulsars are thought to be highly magnetized neutron stars (NSs). The crustal magnetic field of a NS usually decays at different timescales in the forms of Hall drift and Ohmic dissipation. The magnetization parameter $\omega_B \tau$ is defined as the ratio of the Ohmic timescale $\tau_{Ohm}$ to the Hall drift timescale $\tau_{Hall}$. During the first several million years, the inner temperature of the newly born neutron star cools from $T = 10^9$ K to $T = 1.0 \times 10^8$ K, and the crustal conductivity increases by three orders of magnitude. In this work, we adopt a unified equations of state for cold non-accreting neutron stars with the Hartree–Fock–Bogoliubov method, developed by Pearson et al. (2018), and choose two fiducial dipole magnetic fields of $B = 1.0 \times 10^{13}$ G and $B = 1.0 \times 10^{14}$ G, four different temperatures, T, and two different impurity concentration parameters, Q, and then calculate the conductivity of the inner crust of NSs and give a general expression of magnetization parameter for young pulsars: $\omega_B \tau \simeq (1 - 50) B_0 / (10^{13}$ G$)$ by using numerical simulations. It was found when $B \leq 10^{15}$ G, due to the quantum effects, the conductivity increases slightly with the increase in the magnetic field, the enhanced magnetic field has a small effect on the matter in the low-density regions of the crust, and almost has no influence the matter in the high-density regions. Then, we apply the general expression of the magnetization parameter to the high braking-index pulsar PSR J1640-4631. By combining the observed arrival time parameters of PSR J1640-4631 with the magnetic induction equation, we estimated the initial rotation period $P_0$, the initial dipole magnetic field $B_0$, the Ohm dissipation timescale $\tau_{Ohm}$ and Hall drift timescale $\tau_{Hall}$. We model the magnetic field evolution and the braking-index evolution of the pulsar and compare the results with its observations. It is expected that the results of this paper can be applied to more young pulsars.

**Keywords:** neutron stars; conductivity; magnetization parameters; ohmic dissipation; hall drift

## 1. Introduction

As one of the most densest stars in the Universe, neutron stars (NSs) are the most important research objects in the field of high-energy astrophysics. Studying NSs can help us understand the properties of matter at extreme circumstances, and a series of extreme physical processes may occur in

the interior of NSs with high density and strong pressure. It is generally recognized that pulsars are thought to be fast-spinning NSs radiating energy from their rotational energy losses. The magnetic field of pulsars is the basis of studying various radiation models and the probe into their internal structures [1], which has received wide attention from researchers. The surface dipole magnetic field is the strongest and has a wide distribution: from $B \sim 10^8$–$10^9$ gauss (G) for millisecond pulsar, through $B \sim 10^{12}$–$10^{13}$ G for normal radio pulsars, to $B \sim 10^{14}$–$10^{15}$ G for magnetars whose radiations are powered by magnetic fields [2–6]. The main method used to determine these magnetic fields is to measure the spin period $P$ of each pulsar and its first derivative $\dot{P}$. Assuming that the pulsar's rotational energy loss is completely dominated by magnetic dipole radiation (MDR), the surface dipole poloidal magnetic field strength at the polar gaps of pulsars, $B_p$, is inferred as

$$P\dot{P} = \frac{2\pi^2 R^6 sin^2\alpha}{3Ic^3} B_p^2 \, , \tag{1}$$

where $I$ and $R$ are the moment of inertia and radius of the star, respectively, $c$ is the speed of light in vacuum, and $\alpha$ is the angle between the rotational axis and the dipole axis (the magnetic inclination angle).

The magnetic field evolution, cooling, and radiation mechanism of pulsars have been studied extensively and deeply in recent years. In order to simulate the cooling process of a NS, we started with a crust at initial temperature $T = 10^9$ K [7,8], a typical value after formation of the crust, at most within hours after birth, and we force the temperature of the isothermal crust to vary according to [9]: $T(t) = 10^9(1 + t_6)^{-1/6}$, $t_6$ is the NS age in $10^6$ years (yrs). If only modified URCA processes are operating [10,11], this approximation is valid during the neutrino-cooling era. This simple approximation is sufficient for capturing the main effect: as the NS's crust cools ($10^9$ K to $10^8$ K in 1 million years), crustal conductivity increases gradually, the Hall drift term is becoming more and more important via the increase in the electron relaxation time. When a significant part of the crustal magnetic field is dissipated and/or it has become much closer a force-free configuration, the field decay continues on a much longer Ohmic timescale [12,13]. In order to study magnetic field dissipation in NS crusts from magnetars to isolated NSs, Pons and Geppert (2007) [9] first performed the long-term simulations of the non-linear magnetic field evolution in realistic NS crusts with a stratified electron number density, $n_e$, and temperature dependent conductivity, $\sigma$. The results show that Ohmic dissipation influenced by Hall drift takes place in NS crusts on a timescale of $\tau_{Ohm} \sim 10^6$ yrs. When the initial magnetic field has magnetar strength, the fast Hall drift results in an initial rapid dissipation stage lasting about $10^4$ yrs, where stable configurations can last for $10^6$ yrs. The effect of Hall drift depends on the initial field strength and structure and how fast the NS cools. During the Hall drift stage, the toroidal field is strongly rearranged and dissipated, after this stage the long-term evolution seems to select, generally, a predominantly quadrupolar/octupolar structure concentrated in the inner crust and which tends to be stronger close to the poles. It is expected that such magnetic rearrangement and relatively rapid decay will produce the observed consequences such as those seen in magnetars, such as giant flares, and outer bursts [2,3].

Previous studies on the Ohmic dissipation mainly focused on the decay of the NS crust magnetic field, which includes calculations of the Ohmic decay eigenmodes in the crust [14], self-similar solutions near the stellar surface [15], Ohmic dissipation equation [16,17] and the dissipation rates of multipole magnetic field [18] . Geppert and Urpin (1994) first studied magnetic field evolution in accreting neutron stars [19]. There are two factors that can slow down the magnetic field decay: the gravitational redshift effect and the intrinsic curved geometry of the spatial section [20]. Due to the effects of general relativity, the timescale of magnetic field decay increases, but keeps in the same order of magnitude in flat spacetime [21–23]. Very recently, Pons and Viganò (2019) [24] reviewed the basic theory describing the magneto-thermal evolution models of NSs, focusing on numerical techniques, and providing a battery of benchmark tests to be used as a reference for present and future code developments. Wang et al. (2019) [25] deduced an eigenvalue equation of Ohmic decay for ordinary NSs under the framework of general relativity, and used the magnetic spot formation and thermoplastic flow

heating model to explain the soft X-ray luminosity of PSR J1640-4631 and its high surface temperature. By calculating the toroidal field decay rate and magnetic energy decay rate, Chen et al. (2019) [26] found that for most of high X-ray luminosity magnetars, the toroidal field Ohmic decay can provide the observed soft X-ray radiations, while for low X-ray luminosity transient magnetars, their soft X-ray radiations may come from rotating energy losses.

In this work, we concentrate on the effects of the cooling of young pulsars (from $T(t) = 10^9$ K to $T(t) = 1 \times 10^8$ K) on the crustal conductivity, and the evolutions of magnetic field and spin-down during the first few million years.

The reminder of this paper is organized as follows: In Section 2, by combining the equation of state and using a practical program, we calculate the conductivity of NS inner curst, and give the range of magnetization parameters; In Section 3, we build a theoretical model, and apply the general expression of magnetization parameter obtained to the high-braking-index pulsar PSR J1640-4631; A sumarry is given in Section 4.

## 2. Electrical Conductivity

### 2.1. Magnetic Induction Equation

In practice, since the distributions of electron number density and electrical conductivity are not constant, the NS crustal magnetic field is more complex and the nonlinear evolution of the Hall term must be considered. Firstly, the induction equation for the evolution of magnetic field in general relativity is given by [13].

$$\frac{\partial \mathbf{B}}{\partial t} = -\nabla \times \left[ \frac{c}{4\pi e n_e} \left[ \nabla \times \left( e^\Phi \mathbf{B} \right) \right] \times \mathbf{B} + \frac{c^2}{4\pi \sigma} \nabla \times \left( e^\Phi \mathbf{B} \right) \right] , \tag{2}$$

where $\Phi$ is gravitational potential, $e^\Phi$ is the relativistic redshift correction ($e^\Phi = Z = (1 - \frac{2GM}{c^2 R})^{1/2}$). $M$ is the neutron star mass, $e$ is the electron charge, $\mathbf{B}$ is the magnetic field, which includes the poloidal component $\mathbf{B}_{pol}$ and the toroidal component $\mathbf{B}_{tor}$, $\sigma$ is the conductivity parallel to the magnetic field. The above equation includes two different effects of the Hall drift and Ohmic dissipation, which act on two distinct timescales. The Hall drift timescale mainly depends on the initial magnetic field $B_0$, the electron number density $n_e$ and the typical magnetic field length-scale $\lambda$.

$$\tau_{Hall} = \frac{4\pi n_e e \lambda^2}{c B_0} = \frac{6.4}{B_{14}} \left( \frac{n_e}{10^{36} \text{ cm}^{-3}} \right) \left( \frac{\lambda}{\text{km}} \right)^2 \times 10^5 \text{ yrs} , \tag{3}$$

where $\lambda$ is approximated as the thickness of the inner crust, that is $\lambda \approx R_c \sim$ (500–800) m, $B_{14}$ is the magnetic field in units of $10^{14}$ G, the Ohmic dissipation timescale is dependent on $\sigma$ and $\lambda$ [25,26],

$$\tau_{Ohm} = \frac{4\pi \sigma \lambda^2}{c^2} = 4.5 \left( \frac{\sigma}{10^{24} \text{ s}^{-1}} \right) \left( \frac{\lambda}{\text{km}} \right)^2 \times 10^6 \text{ yrs} . \tag{4}$$

Inserting some typical numbers of $\sigma$, $\lambda$ and $n_e$, we get the Ohmic dissipation timescale $\tau_{Ohm} \sim 10^6$ yrs, and the Hall drift timesacle $\tau_{Hall} \sim \left( 10^4 \sim 10^5 \right)$ yrs [27–29], the Hall timescale is usually 1–2 orders of magnitude lower than the Ohmic timescale. The Hall drift term of Equation (3) is a consequence of the Lorentz force acting on the electrons. The tensor components of the electric conductivity are derived in the relaxation time approximation [30]. From Equations (3) and (4), we get the magnetization parameter:

$$\omega_B \tau \equiv \frac{\tau_{Ohm}}{\tau_{Hall}} = \frac{\sigma B_0}{n_e e c} . \tag{5}$$

Then, the first term on the right-hand side of Equation (3) becomes $\frac{c^2}{4\pi} \nabla \times \omega_B \tau [(\nabla \times \mathbf{B}) \times \mathbf{b}]$, here $\mathbf{b} = \mathbf{B}/B$ is the unit vector in the direction of the magnetic field $\mathbf{B}$, and $B$ the magnetic field

strength. It is obvious that the Hall term is proportional to the magnetization parameter $\omega_B \tau$. If the value of $\omega_B \tau$ significantly exceeds unity, the Hall drift dominates, which results in a very different field evolution from the pure Ohmic case, and the electric conductivity perpendicular to the magnetic field will be suppressed by a large magnetization parameter [29]. For a typical NS with $B_0 = 10^{14}$ G, the magnetic field decay is initially dominated by the Hall drift, followed by pure Ohmic decay after proceeding on a timescale on the order of $10^6$ years.

Depending on the strength and structure of the initial magnetic field $B_0$, the Hall drift phase lasts for a few $10^3$–$10^4$ yrs, characterized by a strong exchange of magnetic field energy between the poloidal component and toroidal component of the field and by the redistribution of magnetic field energy between different timescales. To further study the relationship between the ratio of Ohmic timescale to Hall timescale and the magnetic field, the conductivity and magnetization parameter of NS crusts must be calculated.

### 2.2. Calculations of Conductivity and Magnetization Parameter

The conductivity of NS crusts determines the magnetization parameter and the magnetic field decay timescales, and the change in conductivity also greatly affects the processes of the magnetic field evolution and the NS cooling. The crustal conductivity is contributed by the electron-phonon scattering and electron-impurity scattering. By definition [31], the conductivity can be expressed as

$$\frac{1}{\sigma} = \frac{1}{\sigma_{hp}} + \frac{1}{\sigma_{imp}}, \tag{6}$$

where $\sigma_{hp}$ and $\sigma_{imp}$ are conductivities due to the electron-phonon scattering and the electron- impurity scattering, respectively. The magnitude of the conductivity depends strongly on temperature $T$ and density $\rho$, the latter spans six or more orders of magnitude. The third parameter determining $\sigma$ is the impurity concentration, which is defined as $\rho = 1/n \times \sum_i n_i (Z - Z_i)^2$, where $n$ is the total ion density and $n_i$ is the density of impurity of the $i$-th species with charge $Z_i$, and $Z$ is the ionic charge in the pure lattice [15]. Since the lower-density regions of the outer crust have a significantly shorter Ohmic decay timescale and will not be included in our model.

On the basis of the previous work [32], Pearson et al. (2018) [33] used nuclear energy-density functional theory to develop a unified treatment of NSs within the framework of the picture of "cold catalysed matter", meaning that thermal, nuclear and beta equilibrium prevail at a temperature $T$ low enough that thermal effects can be neglected for the composition and pressure. By making use of the latest experimental nuclear mass data [34] and applying the Hartree–Fock–Bogoliubov (HFB) calculation method of nuclear interactions between two-body and three-body, they gave a set of nuclear mass models HFB-22, HFB-23, HFB-24, HFB-25 and HFB-26, corresponding to the energy-density functionals for BSk22, BSk23, BSk24, BSk25 and BSk26, respectively. In this way, each functional was used to calculate the composition, pressure density relationship and chemical potential of neutron stars, and study the influence of the uncertainty of symmetry energy on the calculation results. The fitting results of Pearson et al. (2018) support the heaviest neutron star that had been observed (the mass of the PSR J1614-2230 is M = 2.01$\pm$ 0.04 M$_\odot$ [35]). The EoS of homogeneous pure neutron matter (NeuM) provided by Li and Schulze (2008) [36] to which they fitted BSk22-25 can be regarded as typically hard, while the APR EoS provided by Akmal, Pandharipande and Ravenhall (1998) [37] to which they fitted BSk26 as typically soft. However, in the inner crust denser region of NSs, the fitting of EoS by BSk22 was uncertain, so it cannot be compared with other functionals. BSk24 and BSk26 have the same symmetric energy coefficient (J = 30 MeV) and the former has a larger value of symmetry-slope coefficient $L$. Since the symmetric energy coefficient (J = 31 MeV) of BSk23 is between BSk22 and BSk24, BSk23 is not considered in this paper. A detailed comparison of the above five functionals was given in Person et al. (2018). In Table 1 we list principle values of computed equilibrium proton number $Z_{eq}$ in the inner crust below proton drip for functionals BSk22, BSk24, BSk25 and BSk26. The crustal baryon number density $n_B$ always can be treated as an invariable quantity even in a strong magnetic

field, the electron fraction is equal to the proton number fraction $Y_p$, then the electron number density calculated as $n_e = Y_e \bar{n} = Z_{eq} \bar{n} / A$ , where $Z_{eq}$ is the computed equilibrium proton number, $\bar{n}$ is the mean baryon number density, and $A$ is the mass number of the nucleus.

**Table 1.** Principal values of $Z_{eq}$ in the inner crust below proton drip for BSk22, BSk24, BSk25 and BSk26. Here $\bar{n}_{min}$ and $\bar{n}_{max}$ are the minimum and maximum baryon number densities, respectively, at which the nuclide is present.

| EoS | $Z_{eq}$ | $\bar{n}_{min}$ (fm$^{-3}$) | $\bar{n}_{max}$ (fm$^{-3}$) |
|-----|----------|-----------------------------|-----------------------------|
| BSk22 | 40 | $2.69 \times 10^{-4}$ | 0.0340 |
| BSk24 | 40 | $2.56 \times 10^{-4}$ | 0.0715 |
| BSk25 | 50 | $2.70 \times 10^{-4}$ | 0.0138 |
| BSk26 | 40 | $2.61 \times 10^{-4}$ | 0.0730 |

In this article, implementing realistic conductivity profiles provided by Potekhin, Pons, and Page (2015) [38] (http://www.ioffe.ru/astro/conduct) into the code and combining with the EoS, we calculate the NS inner crustal conductivity. Since the effect of a strong magnetic field on the electrical conductivity was taking into account in reference [38], our results will be more reliable and will better reflect the actual situation of the NS crustal conductivity, compared with previous calculations.

Since the lattice phonons restrict the motion of electrons, the heat and charge transports are dominated by electron-phonon scattering at higher temperatures and lower densities, whereas heat and charge transport are dominated by the electron-impurity scattering at higher densities. In references [29,31], the authors studied the low-density crust dominated by electron-phonon scattering and gave a range of impurity parameter $Q \sim 10^{-4}$–$10^{-2}$, recently, studies of the high-density layers dominated by electron-impurity scattering suggested higher impurity parameters $Q \geq 1$ [25–27].

A newly born NS is very hot, its crustal temperature can be high as $10^9$ K, and the neutrino radiation cooling dominates. Firstly, we choose a crust with a fiducial magnetic field $B = 1.0 \times 10^{13}$ G, and two different impurity parameters $Q = 0.01$ and $Q = 1.0$, when the temperature cools from $10^9$ K to $5 \times 10^8$ K, then calculate partial values of the conductivity $\sigma$ and and magnetized parameter $\omega_B \tau$. Because of the suppression of the conductivity in the direction orthogonal to a strong magnetic field, here we only consider the conductivity parallel to the direction of the magnetic field. The unit of magnetization parameter $\omega_B \tau$ is in units of $B_0 / (10^{13}\,\text{G})$, and the crust is thought to be isothermal. The calculated results are listed in Table 2.

It is found that (1) $\sigma$ and $\omega_B \tau$ increase with the increase in $\rho$ when $T$, $Q$ and $B_p$ are given; (2) $\sigma$ and $\omega_B \tau$ decrease with the increase in $Q$ when $T$, $\rho$ and $B_p$ are given; (3) $\sigma$ and $\omega_B \tau$ decrease with the increase in $T$ when $Q$, $\rho$ and $B_p$ are given. We compare the results in the case of $Q = 0.01$ with those of $Q = 1.0$, and find that both the differences between conductivities and the differences between magnetization parameters are very small, and the magnetization parameters are distributed in a range of $\omega_B \tau \sim (10^{-1} - 3) B_0 / (10^{13}\,\text{G})$.

Keeping the fiducial magnetic field constant, when the crustal temperature drops to $T = 1.0 \times 10^8$ K, we assume $Q = 0.01$ and $Q = 1.0$, then obtain the magnetization parameter's ranges $\omega_B \tau \sim (0.5 - 3) B_0 / (10^{13}\,\text{G})$ and $\omega_B \tau \sim (0.5 - 2) B_0 / (10^{13}\,\text{G})$, respectively. Table 3 lists partial values of $\sigma$ and $\omega_B \tau$. From Table 3, one can see that the conductivity increases over one order of magnitude when we keep constant $Q$ and $\rho$ and let $T$ drops by one order of magnitude. Keeping the fiducial magnetic field constant, when the crustal temperature drops to $T = 1.0 \times 10^8$ K, we assume $Q = 0.01$ and $Q = 1.0$, and obtain the magnetization parameter's ranges $\omega_B \tau \sim (4.9 - 51) B_0 / (10^{13}\,\text{G})$ and $\omega_B \tau \sim (4.6 - 10) B_0 / (10^{13}\,\text{G})$, respectively. Table 3 lists partial values of $\sigma$ and $\omega_B \tau$.

**Table 2.** Partial values of electrical conductivity and magnetization parameter for two different temperatures T and two different impurity parameters $Q$ in the inner crust of NSs for the nuclear mass models HFB-22, HFB-24 and HFB-26. The unit of magnetization parameter $\omega_B \tau$ is the normalized magnetic field $B_0/(10^{13}\,\mathrm{G})$ when the dipolar magnetic field strength $B_0 = 1.0 \times 10^{13}$ G. Here the crust is assumed to be isothermal.

| | | | | 1.0e8 K | | | | 1.0e9 K | | | |
|---|---|---|---|---|---|---|---|---|---|---|---|
| | | | | $Q = 0.01$ | | $Q = 1$ | | $Q = 0.01$ | | $Q = 1$ | |
| $\bar{n}_b$ (fm$^{-3}$) | $\rho$ (g cm$^{-3}$) | $Y_e$ | $n_e$ (cm$^{-3}$) | $\sigma$ (s$^{-1}$) | $\omega_B\tau$ | $\sigma$ (s$^{-1}$) | $\omega_B\tau$ | $\sigma$ (s$^{-1}$) | $\omega_B\tau$ | $\sigma$ (s$^{-1}$) | $\omega_B\tau$ |
| | | | | HFB-22 | | | | | | | |
| 2.700e-04 | 4.513e11 | 2.955e-01 | 7.979e34 | 6.56e22 | 0.571 | 6.51e22 | 0.567 | 3.52e22 | 0.306 | 3.50e22 | 0.305 |
| 5.253e-04 | 8.790e11 | 1.839e-01 | 9.658e34 | 7.66e22 | 0.551 | 7.60e22 | 0.546 | 4.05e22 | 0.291 | 4.03e22 | 0.290 |
| 8.778e-04 | 1.470e12 | 1.294e-01 | 1.136e35 | 8.82e22 | 0.539 | 8.75e22 | 0.535 | 4.60e22 | 0.281 | 4.58e22 | 0.280 |
| 1.194e-03 | 2.000e12 | 1.058e-01 | 1.263e35 | 9.72e22 | 0.534 | 9.63e22 | 0.529 | 5.02e22 | 0.276 | 5.00e22 | 0.275 |
| 1.593e-03 | 2.670e12 | 8.832e-02 | 1.407e35 | 1.08e23 | 0.533 | 1.06e23 | 0.523 | 5.51e22 | 0.272 | 5.48e22 | 0.270 |
| 2.707e-03 | 4.540e12 | 6.510e-02 | 1.762e35 | 1.34e23 | 0.528 | 1.33e23 | 0.524 | 6.73e22 | 0.265 | 6.69e22 | 0.264 |
| 3.726e-03 | 6.250e12 | 5.526e-02 | 2.059e35 | 1.58e23 | 0.533 | 1.56e23 | 0.526 | 7.80e22 | 0.263 | 7.75e22 | 0.261 |
| 4.994e-03 | 8.380e12 | 4.826e-02 | 2.410e35 | 1.88e23 | 0.542 | 1.85e23 | 0.533 | 9.11e22 | 0.262 | 9.05e22 | 0.260 |
| 8.931e-03 | 1.500e13 | 3.841e-02 | 3.431e35 | 2.86e23 | 0.579 | 2.81e23 | 0.569 | 1.33e23 | 0.269 | 1.32e23 | 0.267 |
| 1.535e-02 | 2.580e13 | 3.221e-02 | 4.944e35 | 4.75e23 | 0.667 | 4.63e23 | 0.650 | 2.11e23 | 0.296 | 2.08e23 | 0.292 |
| 2.800e-02 | 4.713e13 | 2.691e-02 | 7.535e35 | 9.95e23 | 0.917 | 9.50e23 | 0.876 | 4.13e23 | 0.381 | 4.05e23 | 0.373 |
| 3.400e-02 | 5.725e13 | 2.562e-02 | 8.709e35 | 1.36e24 | 1.084 | 1.28e24 | 1.021 | 5.51e23 | 0.439 | 5.38e23 | 0.429 |
| | | | | HFB-24 | | | | | | | |
| 2.570e-04 | 4.296e11 | 3.028e-01 | 7.783e34 | 6.44e22 | 0.575 | 6.39e22 | 0.570 | 3.46e22 | 0.309 | 3.44e22 | 0.307 |
| 2.788e-04 | 4.660e11 | 2.859e-01 | 7.970e34 | 6.56e22 | 0.572 | 6.51e22 | 0.567 | 3.52e22 | 0.307 | 3.50e22 | 0.305 |
| 5.253e-04 | 8.790e11 | 1.847e-01 | 9.703e34 | 7.70e22 | 0.551 | 7.63e22 | 0.546 | 4.06e22 | 0.291 | 4.05e22 | 0.290 |
| 8.778e-04 | 1.470e12 | 1.325e-01 | 1.163e35 | 9.01e22 | 0.538 | 8.93e22 | 0.533 | 4.69e22 | 0.280 | 4.66e22 | 0.278 |
| 1.194e-03 | 2.000e12 | 1.100e-01 | 1.314e35 | 1.01e23 | 0.534 | 9.97e22 | 0.527 | 5.18e22 | 0.274 | 5.15e22 | 0.272 |
| 2.093e-03 | 3.510e12 | 8.114e-02 | 1.699e35 | 1.29e23 | 0.537 | 1.28e23 | 0.533 | 6.47e22 | 0.269 | 6.44e22 | 0.268 |
| 2.707e-03 | 4.540e12 | 7.186e-02 | 1.945e35 | 1.48e23 | 0.528 | 1.46e23 | 0.521 | 7.33e22 | 0.262 | 7.29e22 | 0.260 |
| 4.991e-03 | 8.380e12 | 5.661e-02 | 2.825e35 | 2.23e23 | 0.548 | 2.20e23 | 0.541 | 1.06e23 | 0.261 | 1.05e23 | 0.258 |
| 8.926e-03 | 1.500e13 | 4.809e-02 | 4.293e35 | 3.77e23 | 0.610 | 3.69e23 | 0.597 | 1.71e23 | 0.277 | 1.69e23 | 0.273 |
| 1.534e-02 | 2.580e13 | 4.289e-02 | 6.578e35 | 7.07e23 | 0.746 | 6.83e23 | 0.721 | 3.03e23 | 0.320 | 2.98e23 | 0.315 |
| 2.778e-02 | 4.680e13 | 3.795e-02 | 1.054e36 | 1.66e24 | 1.094 | 1.55e24 | 1.021 | 6.64e23 | 0.437 | 6.46e23 | 0.426 |
| 3.000e-02 | 5.055e13 | 3.729e-02 | 1.119e36 | 1.87e24 | 1.161 | 1.74e24 | 1.080 | 7.44e23 | 0.462 | 7.23e23 | 0.449 |
| 5.000e-02 | 8.437e13 | 3.328e-02 | 1.664e36 | 4.93e24 | 2.057 | 4.27e24 | 1.782 | 1.83e24 | 0.764 | 1.73e24 | 0.722 |
| 5.634e-02 * | 9.510e13 * | 3.279e-02 | 1.847e36 | 6.65e24 | 2.500 | 5.57e24 | 2.094 | 2.41e24 | 0.906 | 2.25e24 | 0.846 |
| | | | | HFB-26 | | | | | | | |
| 2.620e-04 | 4.379e11 | 2.996e-01 | 7.850e34 | 6.47e22 | 0.572 | 6.42e22 | 0.568 | 3.47e22 | 0.307 | 3.46e22 | 0.306 |
| 2.788e-04 | 4.660e11 | 2.866e-01 | 7.988e34 | 6.56e22 | 0.570 | 6.51e22 | 0.566 | 3.52e22 | 0.306 | 3.50e22 | 0.304 |
| 5.252e-04 | 8.790e11 | 1.833e-01 | 9.629e34 | 7.64e22 | 0.551 | 7.57e22 | 0.546 | 4.03e22 | 0.291 | 4.02e22 | 0.290 |
| 8.777e-04 | 1.470e12 | 1.298e-01 | 1.139e35 | 8.83e22 | 0.538 | 8.75e22 | 0.533 | 4.60e22 | 0.280 | 4.58e22 | 0.279 |
| 1.194e-03 | 2.000e12 | 1.066e-01 | 1.273e35 | 9.76e22 | 0.532 | 9.67e22 | 0.528 | 5.04e22 | 0.275 | 5.01e22 | 0.273 |
| 1.593e-03 | 2.670e12 | 8.955e-02 | 1.427e35 | 1.09e23 | 0.530 | 1.07e23 | 0.521 | 5.54e22 | 0.270 | 5.51e22 | 0.268 |
| 2.094e-03 | 3.510e12 | 7.668e-02 | 1.605e35 | 1.22e23 | 0.528 | 1.20e23 | 0.519 | 6.14e22 | 0.266 | 6.11e22 | 0.264 |
| 2.707e-03 | 4.540e12 | 6.698e-02 | 1.813e35 | 1.37e23 | 0.525 | 1.36e23 | 0.521 | 6.86e22 | 0.263 | 6.82e22 | 0.261 |
| 4.993e-03 | 8.380e12 | 5.098e-02 | 2.546e35 | 1.97e23 | 0.537 | 1.95e23 | 0.532 | 9.51e22 | 0.259 | 9.44e22 | 0.257 |
| 8.929e-03 | 1.500e13 | 4.224e-02 | 3.772e35 | 3.16e23 | 0.582 | 3.10e23 | 0.571 | 1.45e23 | 0.267 | 1.44e23 | 0.265 |
| 1.534e-02 | 2.580e13 | 3.758e-02 | 5.765e35 | 5.70e23 | 0.687 | 5.53e23 | 0.666 | 2.47e23 | 0.298 | 2.44e23 | 0.294 |
| 2.778e-02 | 4.680e13 | 3.456e-02 | 9.601e35 | 1.36e24 | 0.984 | 1.28e24 | 0.926 | 5.41e23 | 0.391 | 5.29e23 | 0.383 |
| 3.000e-02 | 5.054e13 | 3.425e-02 | 1.028e36 | 1.55e24 | 1.047 | 1.45e24 | 0.980 | 6.11e23 | 0.413 | 5.96e23 | 0.403 |
| 5.000e-02 | 8.437e13 | 3.278e-02 | 1.639e36 | 4.44e24 | 1.881 | 3.89e24 | 1.648 | 1.62e24 | 0.686 | 1.54e24 | 0.652 |
| 5.634e-02 * | 9.510e13 * | 3.274e-02 | 1.845e36 | 6.08e24 | 2.288 | 5.15e24 | 1.938 | 2.16e24 | 0.813 | 2.03e24 | 0.764 |

* The sign denotes that the computed equilibrium proton number $Z_{eq}$ begins to deviate from a standard value of $Z = 40$.

**Table 3.** Partial values of electrical conductivity and magnetization parameter for two different temperatures T and two different impurity parameters $Q$ in the inner crust of NSs for the nuclear mass models HFB-22, HFB-24 and HFB-26. The unit of magnetization parameter $\omega_B\tau$ is the normalized magnetic field $B_0/(10^{13}\,\text{G})$ when the dipolar magnetic field strength $B_0 = 1.0 \times 10^{13}\,\text{G}$. Here the crust is assumed to be isothermal.

| | | | | 1.0e7 K | | | | 1.0e8 K | | | |
|---|---|---|---|---|---|---|---|---|---|---|---|
| | | | | Q = 0.01 | | Q = 1 | | Q = 0.01 | | Q = 1 | |
| $\bar{n}_b$ (fm$^{-3}$) | $\rho$ (g cm$^{-3}$) | $Y_e$ | $n_e$ (cm$^{-3}$) | $\sigma$ (s$^{-1}$) | $\omega_B\tau$ | $\sigma$ (s$^{-1}$) | $\omega_B\tau$ | $\sigma$ (s$^{-1}$) | $\omega_B\tau$ | $\sigma$ (s$^{-1}$) | $\omega_B\tau$ |
| | | | | HFB-22 | | | | | | | |
| 2.700e-04 | 4.513e11 | 2.955e-01 | 7.979e34 | 4.77e25 | 415.15 | 7.38e24 | 64.23 | 5.64e23 | 4.91 | 5.30e23 | 4.61 |
| 5.253e-04 | 8.790e11 | 1.839e-01 | 9.658e34 | 6.19e25 | 445.08 | 8.12e24 | 58.39 | 7.00e23 | 5.03 | 6.51e23 | 4.68 |
| 8.778e-04 | 1.470e12 | 1.294e-01 | 1.136e35 | 7.65e25 | 467.65 | 8.77e24 | 53.61 | 8.51e23 | 5.20 | 7.84e23 | 4.79 |
| 1.194e-03 | 2.000e12 | 1.058e-01 | 1.263e35 | 8.81e25 | 484.41 | 9.21e24 | 50.64 | 9.71e23 | 5.34 | 8.87e23 | 4.88 |
| 1.593e-03 | 2.670e12 | 8.832e-02 | 1.407e35 | 1.02e26 | 503.44 | 9.68e24 | 47.78 | 1.11e24 | 5.48 | 1.01e24 | 4.98 |
| 2.094e-03 | 3.510e12 | 7.511e-02 | 1.573e35 | 1.18e26 | 520.94 | 1.02e25 | 45.03 | 1.28e24 | 5.65 | 1.15e24 | 5.08 |
| 3.726e-03 | 6.250e12 | 5.526e-02 | 2.059e35 | 1.72e26 | 580.11 | 1.15e25 | 38.79 | 1.83e24 | 6.17 | 1.59e24 | 5.36 |
| 4.994e-03 | 8.380e12 | 4.826e-02 | 2.410e35 | 2.16e26 | 622.41 | 1.23e25 | 35.44 | 2.28e24 | 6.57 | 1.94e24 | 5.59 |
| 8.931e-03 | 1.500e13 | 3.841e-02 | 3.431e35 | 3.74e26 | 756.99 | 1.44e25 | 29.15 | 3.89e24 | 7.87 | 3.09e24 | 6.25 |
| 1.535e-02 | 2.580e13 | 3.221e-02 | 4.944e35 | 7.06e26 | 991.66 | 1.70e25 | 23.87 | 7.24e24 | 10.17 | 5.11e24 | 7.18 |
| 2.781e-02 | 4.680e13 | 2.696e-02 | 7.498e35 | 1.66e27 | 1537.4 | 2.08e25 | 19.26 | 1.69e25 | 15.65 | 9.36e24 | 8.67 |
| 3.400e-02 | 5.725e13 | 2.562e-02 | 8.709e35 | 2.39e27 | 1905.8 | 2.25e25 | 17.94 | 2.43e25 | 19.38 | 1.17e25 | 9.33 |
| | | | | HFB-24 | | | | | | | |
| 2.570e-04 | 4.296e11 | 3.028e-01 | 7.783e34 | 4.68e25 | 417.58 | 7.31e24 | 65.22 | 5.53e23 | 4.93 | 5.20e23 | 4.64 |
| 2.788e-04 | 4.660e11 | 2.859e-01 | 7.970e34 | 4.81e25 | 419.11 | 7.39e24 | 64.39 | 5.67e23 | 4.94 | 5.33e23 | 4.64 |
| 5.253e-04 | 8.790e11 | 1.847e-01 | 9.703e34 | 6.28e25 | 449.46 | 8.14e24 | 58.26 | 7.08e23 | 5.07 | 6.58e23 | 4.71 |
| 8.778e-04 | 1.470e12 | 1.325e-01 | 1.163e35 | 7.97e25 | 475.90 | 8.87e24 | 52.96 | 8.82e23 | 5.27 | 8.10e23 | 4.84 |
| 1.194e-03 | 2.000e12 | 1.100e-01 | 1.314e35 | 9.37e25 | 495.20 | 9.39e24 | 49.63 | 1.03e24 | 5.44 | 9.35e23 | 4.94 |
| 2.093e-03 | 3.510e12 | 8.114e-02 | 1.699e35 | 1.33e26 | 543.62 | 1.05e25 | 42.92 | 1.43e24 | 5.84 | 1.27e24 | 5.19 |
| 2.707e-03 | 4.540e12 | 7.186e-02 | 1.945e35 | 1.60e26 | 571.27 | 1.12e25 | 39.99 | 1.71e24 | 6.11 | 1.50e24 | 5.36 |
| 3.724e-03 | 6.250e12 | 6.288e-02 | 2.342e35 | 2.09e26 | 619.72 | 1.22e25 | 36.18 | 2.20e24 | 6.52 | 1.88e24 | 5.57 |
| 4.991e-03 | 8.380e12 | 5.661e-02 | 2.825e35 | 2.76e26 | 678.47 | 1.32e25 | 32.45 | 2.88e24 | 7.08 | 2.39e24 | 5.88 |
| 8.926e-03 | 1.500e13 | 4.809e-02 | 4.293e35 | 5.33e26 | 862.19 | 1.59e25 | 25.72 | 5.49e24 | 8.88 | 4.11e24 | 6.65 |
| 1.534e-02 | 2.580e13 | 4.289e-02 | 6.578e35 | 1.13e27 | 1192.9 | 1.92e25 | 20.27 | 1.15e25 | 12.14 | 7.25e24 | 7.65 |
| 2.778e-02 | 4.680e13 | 3.795e-02 | 1.054e36 | 2.96e27 | 1950.2 | 2.42e25 | 15.94 | 3.00e25 | 19.77 | 1.35e25 | 8.89 |
| 3.000e-02 | 5.055e13 | 3.729e-02 | 1.119e36 | 3.38e27 | 2097.6 | 2.50e25 | 15.51 | 3.44e25 | 21.35 | 1.45e25 | 9.00 |
| 5.000e-02 | 8.437e13 | 3.328e-02 | 1.664e36 | 9.59e27 | 4002.2 | 3.19e25 | 13.31 | 9.85e25 | 41.11 | 2.41e25 | 10.10 |
| 5.634e-02 * | 9.510e13 * | 3.279e-02 | 1.847e36 | 1.32e28 | 4963.0 | 3.43e25 | 12.90 | 1.36e26 | 51.13 | 2.74e25 | 10.30 |
| | | | | HFB-26 | | | | | | | |
| 2.620e-04 | 4.379e11 | 2.996e-01 | 7.850e34 | 4.70e25 | 415.78 | 7.33e24 | 64.84 | 5.55e23 | 4.91 | 5.22e23 | 4.62 |
| 2.788e-04 | 4.660e11 | 2.866e-01 | 7.988e34 | 4.80e25 | 417.29 | 7.39e24 | 64.25 | 5.66e23 | 4.92 | 5.32e23 | 4.62 |
| 5.252e-04 | 8.790e11 | 1.833e-01 | 9.629e34 | 6.22e25 | 448.59 | 8.11e24 | 58.49 | 7.01e23 | 5.06 | 6.52e23 | 4.70 |
| 8.777e-04 | 1.470e12 | 1.298e-01 | 1.139e35 | 7.74e25 | 471.91 | 8.78e24 | 53.53 | 8.59e23 | 5.24 | 7.90e23 | 4.82 |
| 1.194e-03 | 2.000e12 | 1.066e-01 | 1.273e35 | 8.97e25 | 489.33 | 9.25e24 | 50.46 | 9.85e23 | 5.37 | 8.99e23 | 4.90 |
| 1.593e-03 | 2.670e12 | 8.955e-02 | 1.427e35 | 1.05e26 | 510.98 | 9.74e24 | 47.40 | 1.14e24 | 5.55 | 1.03e24 | 5.01 |
| 2.707e-03 | 4.540e12 | 6.698e-02 | 1.813e35 | 1.45e26 | 555.40 | 1.09e25 | 41.75 | 1.55e24 | 5.94 | 1.37e24 | 5.25 |
| 3.726e-03 | 6.250e12 | 5.756e-02 | 2.144e35 | 1.84e26 | 595.98 | 1.17e25 | 37.90 | 1.95e24 | 6.32 | 1.68e24 | 5.44 |
| 4.993e-03 | 8.380e12 | 5.098e-02 | 2.546e35 | 2.36e26 | 643.71 | 1.26e25 | 34.37 | 2.48e24 | 6.76 | 2.09e24 | 5.70 |
| 8.929e-03 | 1.500e13 | 4.224e-02 | 3.772e35 | 4.33e26 | 797.18 | 1.50e25 | 27.62 | 4.47e24 | 8.23 | 3.47e24 | 6.39 |
| 1.534e-02 | 2.580e13 | 3.758e-02 | 5.765e35 | 8.95e26 | 1078.1 | 1.80e25 | 21.68 | 9.13e24 | 11.00 | 6.10e24 | 7.35 |
| 2.778e-02 | 4.680e13 | 3.456e-02 | 9.601e35 | 2.44e27 | 1764.9 | 2.29e25 | 16.56 | 2.48e25 | 17.94 | 1.20e25 | 8.68 |
| 3.000e-02 | 5.054e13 | 3.425e-02 | 1.028e36 | 2.83e27 | 1911.7 | 2.37e25 | 16.01 | 2.87e25 | 19.39 | 1.31e25 | 8.85 |
| 5.000e-02 | 8.437e13 | 3.278e-02 | 1.639e36 | 8.88e27 | 3762.4 | 3.09e25 | 13.09 | 9.10e25 | 38.56 | 2.31e25 | 9.79 |
| 5.634e-02 * | 9.510e13 * | 3.274e-02 | 1.845e36 | 1.23e28 | 4629.6 | 3.34e25 | 12.57 | 1.27e26 | 47.80 | 2.65e25 | 9.97 |

\* The sign denotes that the computed equilibrium proton number $Z_{eq}$ begins to deviate from a standard value of $Z = 40$.

From Table 3, one can see that the conductivity increases over 1–2 order of magnitude when we keep constant $Q$ and $\rho$ and let $T$ drops by one order of magnitude. Table 4 lists partial values of $\sigma$ and $\omega_B\tau$ for the nuclear mass model HFB-25, we compare the results in the case of $Q = 0.01$ with those of $Q = 1.0$, and find that both the differences between conductivities and the differences between magnetization parameters are very small, and the magnetization parameters are distributed in a range of $\omega_B\tau \sim (4.0\text{–}10.0)B_0/(10^{13}\,\text{G})$ and $\omega_B\tau \sim (4.0\text{–}7.0)B_0/(10^{13}\,\text{G})$. If $\rho > 9.510 \times 10^{13}\,\text{g cm}^{-3}$, the calculated equilibrium number of protons, as shown in the third column of Table 5, will deviate from a standard value of $Z = 40$. Finally, we assume a fiducial magnetic field $B_0 = 1.0 \times 10^{14}\,\text{G}$,

and repeat the above calculations, and list partial results in Tables A1 and A2 in Appendix A. It is found that when $B \leq 10^{15}$ G, due to the quantum effects, the conductivity increases slightly with the increase in the magnetic field, the enhanced magnetic field has a small effect on the matter in the low-density regions of the crust, and almost has no influence the matter in the high-density regions. Note that, this weak dependence of the longitudinal conductivity on the magnetic field is not absolute, it applies only to relatively large densities and not too strong magnetic fields. If the field strength is far larger than $10^{15}$ G, there will be significant quantum effects of conductivity, such as oscillations around the classical value.

**Table 4.** Same as in Table 3, for the nuclear mass model HFB-25 (Cited from Person et al. (2008).

| | | | | 1.0e7 K | | | | 1.0e8 K | | | |
|---|---|---|---|---|---|---|---|---|---|---|---|
| | | | | Q = 0.01 | | Q = 1 | | Q = 0.01 | | Q = 1 | |
| $\bar{n}_b$ (fm$^{-3}$) | $\rho$ (g cm$^{-3}$) | $Z_{eq}$ | $n_e$ (cm$^{-3}$) | $\sigma$ (s$^{-1}$) | $\omega_B\tau$ | $\sigma$ (s$^{-1}$) | $\omega_B\tau$ | $\sigma$ (s$^{-1}$) | $\omega_B\tau$ | $\sigma$ (s$^{-1}$) | $\omega_B\tau$ |
| 2.800e-04 | 4.681e11 | 49.99 | 7.970e34 | 4.24e25 | 369.4 | 8.68e24 | 75.63 | 5.05e23 | 4.40 | 4.82e23 | 4.20 |
| 5.000e-04 | 8.366e11 | 49.99 | 9.629e34 | 5.47e25 | 394.5 | 9.62e24 | 69.38 | 6.26e23 | 4.51 | 5.94e23 | 4.28 |
| 1.000e-03 | 1.675e12 | 49.99 | 1.258e35 | 7.82e25 | 431.7 | 1.10e25 | 60.72 | 8.70e23 | 4.80 | 8.15e23 | 4.50 |
| 5.000e-03 | 8.397e12 | 49.99 | 3.162e35 | 3.04e26 | 667.7 | 1.71e25 | 37.56 | 3.19e24 | 7.01 | 2.71e24 | 5.95 |
| 1.000e-02 | 1.682e13 | 49.99 | 5.464e35 | 7.70e26 | 978.6 | 2.19e25 | 27.83 | 7.92e24 | 10.07 | 5.86e24 | 7.45 |

**Table 5.** Comparison of the nuclear mass models HFB-24 and HFB-26 in the higher matter-density layers. The third column denotes the computed equilibrium proton number, and the other parameters are the same as in Table 3.

| | | | | 1.0e7 K | | | | 1.0e8 K | | | |
|---|---|---|---|---|---|---|---|---|---|---|---|
| | | | | Q = 0.01 | | Q = 1 | | Q = 0.01 | | Q = 1 | |
| $\bar{n}_b$ (fm$^{-3}$) | $\rho$ (g cm$^{-3}$) | $Z_{eq}$ | $n_e$ (cm$^{-3}$) | $\sigma$ (s$^{-1}$) | $\omega_B\tau$ | $\sigma$ (s$^{-1}$) | $\omega_B\tau$ | $\sigma$ (s$^{-1}$) | $\omega_B\tau$ | $\sigma$ (s$^{-1}$) | $\omega_B\tau$ |
| | | | | HFB-24 | | | | | | | |
| 5.000e-02 | 8.437e13 | 39.91 | 1.664e36 | 9.59e27 | 4002.2 | 3.19e25 | 13.31 | 9.85e25 | 41.11 | 2.41e25 | 10.10 |
| 5.634e-02 | 9.510e13 | 39.80 | 1.847e36 | 1.32e28 | 4963.0 | 3.43e25 | 12.90 | 1.36e26 | 51.13 | 2.74e25 | 10.30 |
| 5.923e-02 | 1.000e14 | 39.65 | 1.936e36 | 1.53e28 | 5488.1 | 3.55e25 | 12.73 | 1.59e26 | 57.03 | 2.91e25 | 10.44 |
| 7.000e-02 | 1.182e14 | 53.54 | 2.297e36 | 3.07e28 | 9281.4 | 5.73e25 | 17.32 | 3.24e26 | 97.95 | 4.88e25 | 14.75 |
| 7.150e-02 | 1.208e14 | 55.26 | 2.350e36 | 3.42e28 | 10106.4 | 6.14e25 | 18.14 | 3.62e26 | 107.0 | 5.26e25 | 15.54 |
| | | | | HFB-26 | | | | | | | |
| 5.000e-02 | 8.437e13 | 39.91 | 1.639e36 | 8.88e27 | 3762.4 | 3.09e25 | 13.09 | 9.10e25 | 38.56 | 2.31e25 | 9.79 |
| 5.634e-02 | 9.510e13 | 39.82 | 1.845e36 | 1.23e28 | 4629.6 | 3.34e25 | 12.57 | 1.27e26 | 47.80 | 2.65e25 | 9.97 |
| 6.513e-02 | 1.100e14 | 39.31 | 2.156e36 | 1.96e28 | 6313.1 | 3.73e25 | 12.01 | 2.05e26 | 66.03 | 3.16e25 | 10.18 |
| 7.000e-02 | 1.183e14 | 38.47 | 2.348e36 | 2.56e28 | 7571.5 | 3.97e25 | 11.74 | 2.71e26 | 80.15 | 3.47e25 | 10.26 |
| 7.300e-02 | 1.233e14 | 37.33 | 2.475e36 | 3.02e28 | 8473.6 | 4.13e25 | 11.59 | 3.23e26 | 90.63 | 3.67e25 | 10.30 |

To vividly describe the changes of conductivity and magnetization parameter with different values of $T$, $\rho$ and $B$, we make diagrams of $\sigma, \omega_B\tau$ versus $\rho$, accoding to the above calculations. As shown in Figure 1, the NS crustal conductivity varies by 3–4 orders of magnitude. A high impurity content could lead to even faster dissipation. Since $B_{13} = B_0/(10^{13}\,\text{G})$ and $B_{14} = B_0/(10^{14}\,\text{G})$, by comparing Figure 1a with Figure 1b, Figure 1c with Figure 1d, Figure 1e with Figure 1f, and Figure 1g with Figure 1h, we do not find significant differences in the simulations with $B = 1 \times 10^{13}$ G and $B = 1 \times 10^{14}$ G. According to the above results, for a strongly magnetized NS, when the crustal temperature drops from $10^9$ K to $1.0 \times 10^8$ K, the ratio of the Ohmic dissipation timescale to the Hall drift timescale is approximately

$$\omega_B\tau = \frac{\tau_{Ohm}}{\tau_{Hall}} = (1 - 50) \times \frac{B_0}{10^{13}\text{G}}. \tag{7}$$

According to Equation (7), the magnetization parameter increases linearly with the initial magnetic field strength $B_0$. To ensure that the Ohmic dissipation timescale is not less than the Hall

drift timescale, and to ensure that our fit is consistent with most models of the magnetic field evolution of NSs, we omit the value of $\omega_B \tau \sim 10^{-1}$.

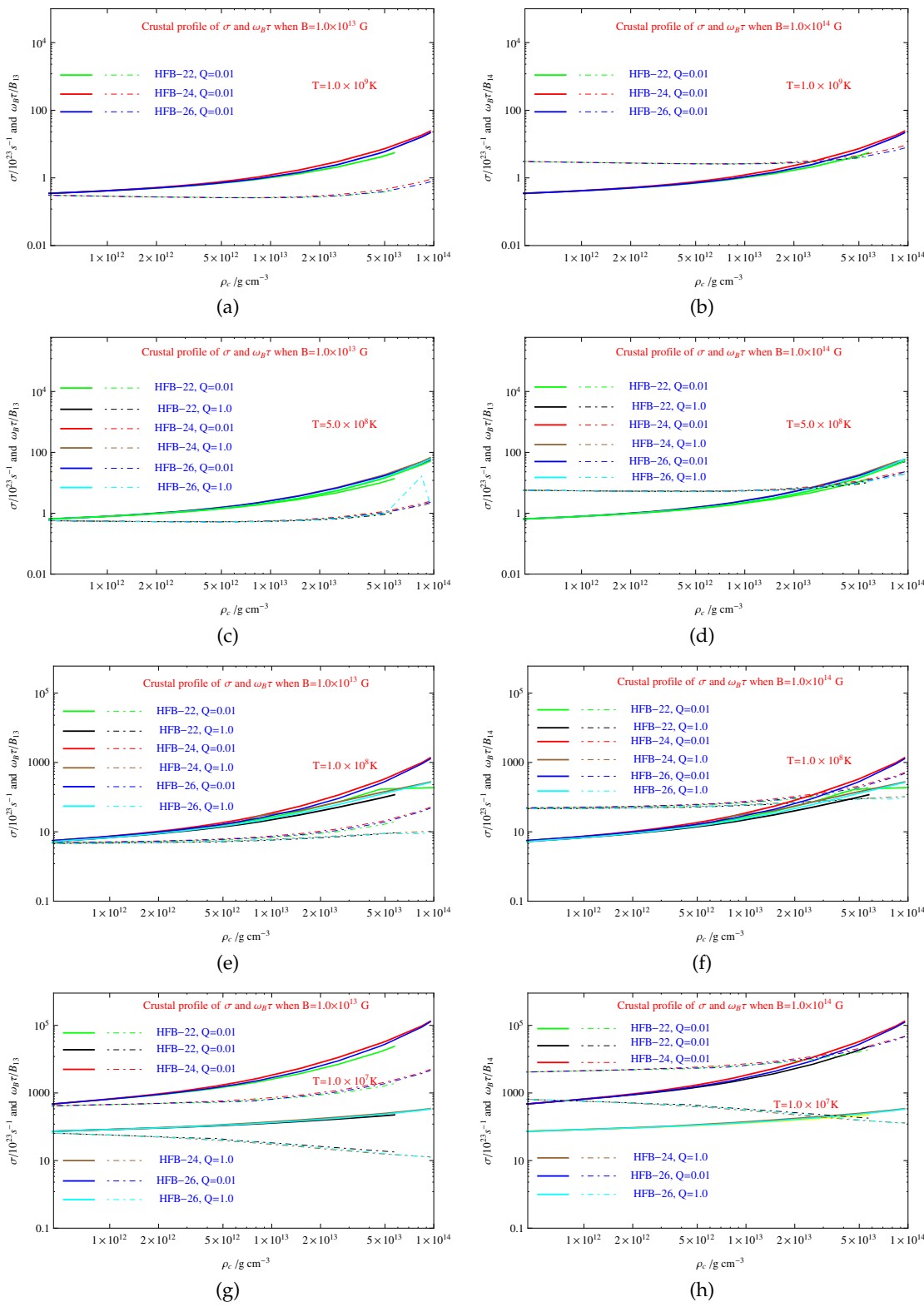

**Figure 1.** The relations of $\sigma$, $\omega_B \tau$ and $\rho$ in the inner crust of a neutron star. The solid line and dot-dashed line are for $\sigma$ and $\omega_B \tau$, respectively. In (**a**,**c**,**e**,**g**), the fiducial magnetic field is $B = 1.0 \times 10^{13}$ G; In (**b**,**d**,**f**,**h**), the fiducial magnetic field is $B = 1.0 \times 10^{14}$ G. The crust is assumed to be isothermal.

It is worth noting that such averaged timescales in Equation (7) are of very restricted use in characterizing the field evolution in NS crusts, since the density vary over many orders of magnitude there. A different issue is whether or not this effect is observable when studying populations of older NSs. Since the onductivity strongly depends on temperature, the Ohmic dissipation timescale increases significantly as the NS crustal temperature cools to $T \leq 10^7$ K (usually $10^7$ years after birth corresponding to $\omega_B \tau \sim 10^{7-8}$ yrs.), and there is no rapid field decay after that age. So in this paper, we want to focus on the evolution of magnetic fields in relatively young pulsars.

## 3. Applying the Magnetization Parameter Expression to the High-Braking-Index Pulsar PSR J1640-4631

### 3.1. The Braking Index and Radiation Characteristics of PSR J1640-4631

Due to the existence of energy loss mechanisms, such as electromagnetic radiation, particle stellar wind, strong neutrino flow and gravitational radiation [39,40], a pulsar spins down. An important and measurable quantity closely related to a pulsar's rotational evolution is the braking index $n$, defined by assuming that the star spins down in the light of a power law

$$\dot{\Omega} = -K\Omega^n \,, \tag{8}$$

where $\Omega$ is the angular velocity, $\dot{\Omega}$ is the first time derivative of $\Omega$, and $K$ is a proportionality constant. According to the standard method, the braking index is defined as

$$n = \frac{\Omega \ddot{\Omega}}{\dot{\Omega}} = \frac{v \ddot{v}}{\dot{v}^2} = 2 - \frac{P \ddot{P}}{\dot{P}^2} \,, \tag{9}$$

where $\ddot{\Omega}$ is the second time derivative of $\Omega$, $v = \Omega/2\pi$ is the spin frequency, $\dot{P}$ and $\ddot{P}$ are the first time derivative and the second time derivative of $P$, respectively. When the magneto-dipole radiation (MDR) solely causes the pulsar to spin down, the braking index is predicted to be $n = 3$. Up to date, only 9 of the $\sim 3000$ known pulsars have reliably measured braking indices, all of which deviate from 3, demonstrating that the spin-down mechanism is not pure MDR. In the case of a varying dipole magnetic field at the pole $B_p$, the braking index $n$ can be simply expressed as follows [13].

$$n = 3 - 4\tau_c \frac{\dot{B}_p}{B_p} \,, \tag{10}$$

where $\tau_c = P/2\dot{P}$ is the characteristic age, $\dot{B}_p$ is the time derivative of $B_p$. For an increasing $B_p$, we will always obtain $n < 3$, owing to an increasing dipole braking torque, whereas $n > 3$ for a decreasing $B_p$.

Recently, PSR J1640-4631, associated with the TeV $\gamma$-ray source HESS J1640-465 was discovered using the *NuSTAR* X-ray observatory [41]. Based on its timing observational data $P = 206.4$ ms, $\dot{P} = 9.7228 \times 10^{-13}$ s s$^{-1}$, and $\ddot{P} = -5.27(13) \times 10^{-24}$ s s$^{-2}$, Archibald et al. [42] derived its braking index to be $n = 3.15(3)$ (here and following all digits in parentheses denote the standard uncertainty), corresponding to its characteristic age $\tau_c = 3550$ yrs. A pulsar's true age $t_{age}$ can be estimated by the age of its associated supernova remnant(SNR), $t_{SNR}$, because it is universally considered that pulsars originate from supernova explosions. G338.3-0.0 is a shell-type SNR and spatially relates to HESS J1640-465, which is considered to be the most luminous $\gamma$-ray source in the Galaxy. The X-ray pulsar PSR J1640-4631 was recently discovered within the shell of SNR G338.3-0.0. Unfortunately, no X-ray emission was detected from the shell of the SNR, thus, the true age of PSR J1640-4631 cannot be estimated from Equation (4) in Reference [12]. The pulsar's true age can be estimated by the following expression [13]

$$t = \frac{P}{(n-1)\dot{P}}[1 - (\frac{P_0}{P})^{n-1}], \quad (n \neq 1)$$

$$t = 2\tau_c \ln(\frac{P}{P_0}), \quad (n = 1). \tag{11}$$

Inserting the values of $P$ and $\dot{P}$ into Equation (11), we estimate the true age of PSR J1640-4631 to be $t_{age} \approx 3130\,\text{yrs}$. The soft X-ray radiation flux $F_X$ [2–10 keV] of PSR J1640-4631 obtained from *Chandra* + *NuStar* telescopes is given by $F_X = 1.8(4) \times 10^{-13}\,\text{erg cm}^{-2}\,\text{s}^{-1}$ [41]. Assuming that the observed X-ray radiation from PSR J1640-4631 is isotropic, the soft X-ray luminosity is estimated as $L_X = 4\pi d F_X = 3.26(72) \times 10^{33}\,\text{erg s}^{-1}$, where $d = 12$ kpc is the distance from G338.3-0.1 to Earth [43]. According to the blackbody thermal radiation, the surface temperature of the star is estimated as $T_s = (L_X/4\pi R^2 \sigma_{SB})^{1/4} \approx 1.54(6) \times 10^6\,\text{K}$, where $\sigma_{SB}$ is the Stefan-Boltzmann constant. This value of $T$ is much higher than $T_s \sim 10^5$ K for common radio pulsars, but it is very near to the observed surface temperatures of magnetars. Recently, Wang et al. (2019) [25] found that observed X-ray flux of the pulsar could be caused by the decay of a multipolar magnetic field near the pole, which is strong enough to activate the slot-gap mechanism. The high surface temperature of the star is attributed either to magnetic spot formation [4] or thermoplastic wave heating due to the decay of the toroidal field near the pole [43].

### 3.2. Theoretical Model of Dipole Magnetic Field Evolution

The maximum NS mass predicted by EoS is model dependent. The largest sample of measured NS masses available for analysis is publicly accessible online at http://www.stellarcollapse.org/, from which one can get a range of about $(1–2)M_\odot$ for the observational NS masses. To date, the relativistic-mean-field (RMF) theory has become a standard method to study nuclear matter and finite-nuclei properties [44–48], but it has not been possible to fit masses in the RMF framework with a precision at all comparable to what was achieved with Skyrme functionals (Pearson et al. (2018) [33]). At high densities the symmetry energy of BSk26 increases much less steeply than that of BSk24, given the much softer EoS of NeuM to which it was fitted (Pearson et al. 2018). In this paper, we choose a medium-mass NS with $M = 1.45 M_\odot$ and $R = 11.5$ km, corresponding to the moment of inertia $I = 1.34(1) \times 10^{45}\,\text{g cm}^2$ for PSR J1640-4631 in the BSK26 EoS.

The evolution of the crustal magnetic field is phenomenologically divided into evolutionary stages: the initial stage with rapid (non-exponential) decay, and a later stage with purely Ohmic dissipation (exponential). For simplicity and for qualitatively investigating the effects of the magnetic field decay, the geometry of the field is assumed to be fixed, and the temporal dependence is included only in the normalized $B_p$ according to

$$B_p(t) = B_0 \frac{\exp(-Zt/\tau_{Ohm})}{1 + \frac{\tau_{Ohm}}{\tau_{Hall}}[1 - \exp(-Zt/\tau_{Ohm})]}, \tag{12}$$

where the effect of general relativity is considered, and the gravitational redshift factor $Z = (1 - \frac{2GM}{c^2 R})^{1/2} \approx 0.9$. The inclusion of Hall drift accelerates the decay of the magnetic field, especially in the early field evolution when $t \ll t_{Ohm}$, during which the Hall term becomes a dominant factor, as given by $B_p \approx B_0(1 + t/\tau_{Hall})^{-1}$. Taking the first derivative of the dipole field with respect to time, we obtain

$$\frac{dB_p}{dt} = \frac{-ZB_p}{\tau_{Ohm}} - \frac{ZB_p^2}{\tau_{Hall}B_0}. \tag{13}$$

If the magnetic field evolution of PSR J1640-4631 cannot be ignored and the dipole braking still dominates, according to Reference [43], the braking law of the pulsar is reformulated as

$$\dot{v}(t) = \frac{2\pi^2 R^6}{3Ic^3} B_p^2(t) v^3(t),$$ (14)

where $B_p(t)$ is determined by Equation (12), and a constant inclination angle $\alpha = 90°$ is assumed for the sake of simplicity. Integrating Equation (14) gives the pulsar's spin frequency

$$v^{-2} = v_0^{-2} + 2\int_0^t \frac{2\pi^2 R^6}{3Ic^3} B_p^2(t')dt'.$$ (15)

From Equation (15), we obtain the relation between the rotational period and time

$$P(t) = [P_0^2 + 2\int_0^t \frac{2\pi^2 R^6 B_0^2}{3Ic^3} \frac{\exp^2(-Zt'/\tau_{Ohm})}{[1 + \frac{\tau_{Ohm}}{\tau_{Hall}}[1 - \exp(-Zt'/\tau_{Ohm})]]^2} dt']^{1/2}.$$ (16)

Let $x = -Zt/\tau_{Ohm}$, then $dt = -\frac{\tau_{Ohm}}{Z}dt$ the second term in Equation (16) becomes

$$\begin{aligned}
\text{Second} \quad &= 2\int_0^t \frac{2\pi^2 R^6 B_0^2}{3Ic^3} \frac{\exp^2(-Zt'/\tau_{Ohm})}{[1 + \omega_B\tau[1 - \exp(-Zt'/\tau_{Ohm})]]^2} dt' \\
&= \frac{4\pi^2 R^6 B_0^2}{3Ic^3} \cdot \frac{-\tau_{Ohm}}{Z(\omega_B\tau)^2} \cdot \int_0^x \frac{e^{2x}}{[1 + \frac{1}{\omega_B\tau} - e^x]^2} dx.
\end{aligned}$$ (17)

Making the following substitutions: $a = 1 + \frac{1}{\omega_B\tau}$ and $k = a - e^x$, then we have $x = ln(a - k)$ and $dx = -\frac{1}{a-k}dk$. Inserting the above substitutions into Equation (17), we get

$$\begin{aligned}
\text{Second} \quad &= \frac{4\pi^2 R^6 B_0^2}{3Ic^3} \cdot \left(\frac{-\tau_{Ohm}}{Z(\omega_B\tau)^2}\right) \cdot \int_{a-1}^{a-\exp(x)} \left(\frac{a-k}{k}\right)^2 \frac{-1}{a-k}dk \\
&= \frac{4\pi^2 R^6 B_0^2}{3Ic^3} \cdot \left(\frac{\tau_{Ohm}}{Z(\omega_B\tau)^2}\right) \cdot \int_{a-1}^{a-\exp(x)} \left(\frac{a}{k^2} - \frac{1}{k}\right) dk \\
&= \frac{4\pi^2 R^6 B_0^2}{3Ic^3} \cdot \left(\frac{\tau_{Ohm}}{Z(\omega_B\tau)^2}\right) \cdot \left[a\left[-\frac{1}{k}\right]_{a-1}^{a-\exp(x)} - [\ln(k)]_{a-1}^{a-\exp(x)}\right] \\
&= \frac{4\pi^2 R^6 B_0^2 \tau_{Ohm}}{3Ic^3 Z(\omega_B\tau)^2} \left[\frac{a}{a-1} - \frac{a}{a-e^x} + \ln(a-1) - \ln(a-e^x)\right],
\end{aligned}$$ (18)

where $k = a - 1$ if $x = 0$. Then Equation (16) is rewritten as

$$P(t) = \left[P_0^2 + \frac{4\pi^2 R^6 B_0^2 \tau_{Ohm}}{3Ic^3 Z(\omega_B\tau)^2} \left[\frac{a}{a-1} - \frac{a}{a-e^x} + \ln(a-1) - \ln(a-e^x)\right]\right]^{1/2}.$$ (19)

For convenience, the period $P(t)$ is denoted as $P(t) = N^{1/2}(t)$,

$$N(t) = \left[P_0^2 + \frac{4\pi^2 R^6 B_0^2 \tau_{Ohm}}{3Ic^3 Z(\omega_B\tau)^2} \left(\frac{a}{a-1} - \frac{a}{a-e^x} + \ln(a-1) - \ln(a-e^x)\right)\right].$$ (20)

Taking the derivative of $P(t)$ with respect to time, we get the time first derivative of the period,

$$\dot{P}(t) = \frac{1}{2} N^{-1/2} \frac{4\pi^2 R^6 B_0^2 \tau_{Ohm}}{3Ic^3 (\omega_B\tau)^2 Z} \left[\ln(a-1) + \frac{a}{a-1} - \frac{a}{a-e^x} - \ln(a-e^x)\right]'.$$ (21)

Taking the derivatives of all the terms in parentheses of Equation (21) with respect to time $t$, we have

$$[\dots]' = \frac{z}{\tau_{Ohm}} \frac{e^{-Zt/\tau_{Ohm}}}{a - e^{-Zt/\tau_{Ohm}}} \left( \frac{a}{a - e^{-Zt/\tau_{om}}} - 1 \right) = \frac{Z}{\tau_{Ohm}} \frac{e^{-2Zt/\tau_{Ohm}}}{\left( a - e^{-Zt/\tau_{Ohm}} \right)^2} . \tag{22}$$

Equation (21) then becomes

$$\dot{P}(t) = \frac{2\pi^2 R^6 B_0^2}{3Ic^3 (\omega_B \tau)^2} \frac{e^{-2Zt/\tau_{Ohm}}}{\left( a - e^{-Zt/\tau_{Ohm}} \right)^2} N^{-1/2} = \frac{2\pi^2 R^6 B_0^2}{3Ic^3 (\omega_B \tau)^2} \frac{e^{-2Zt/\tau_{Ohm}}}{\left( a - e^{-Zt/\tau_{Ohm}} \right)^2} P(t)^{-1} . \tag{23}$$

Similarly, taking the derivative of $\dot{P}(t)$ with respect to time, the second derivative of the period can be expressed as

$$
\begin{aligned}
\ddot{P}(t) &= - \left( \frac{2\pi^2 R^6 B_0^2}{3Ic^3 (\omega_B \tau)^2} \right)^2 \frac{e^{-4Zt/\tau_{Ohm}} N^{-3/2}}{\left( a - e^{-Zt/\tau_{Ohm}} \right)^4} - \frac{4\pi^2 R^6 B_0^2 Z}{3Ic^3 (\omega_B \tau)^2 \tau_{Ohm}} \frac{a e^{-2Zt/\tau_{Ohm}} N^{-1/2}}{\left( a - e^{-Zt/\tau_{Ohm}} \right)^3} \\
&= - \left[ \frac{2\pi^2 R^6 B_0^2}{3Ic^3 (\omega_B \tau)^2} \right]^2 \frac{e^{-4Zt/\tau_{Ohm}}}{\left( a - e^{-Zt/\tau_{Ohm}} \right)^4 P(t)^3} - \frac{4\pi^2 R^6 B_0^2 Z}{3Ic^3 (\omega_B \tau)^2 \tau_{Ohm}} \frac{a e^{-2Zt/\tau_{Ohm}}}{\left( a - e^{-Zt/\tau_{Ohm}} \right)^3 P(t)} . \tag{24}
\end{aligned}
$$

From the expression of $\dot{P}(t)$, we obtain

$$\frac{2\pi^2 R^6 B_0^2}{3Ic^3 (\omega_B \tau)^2} = \dot{P}(t) P(t) \left( a - e^{-Zt/\tau_{Ohm}} \right)^2 e^{2Zt/\tau_{Ohm}} . \tag{25}$$

Plugging the expression of $\ddot{P}(t)$ into Equation (25), we further simplify Equation (25) as

$$
\begin{aligned}
\ddot{P}(t) &= - \left[ \dot{P}(t) P(t) \left( a - e^{-Zt/\tau_{Ohm}} \right)^2 e^{2Zt/\tau_{Ohm}} \right]^2 \times \frac{e^{-4Zt/\tau_{Ohm}}}{\left( a - e^{-2Zt/\tau_{Ohm}} \right)^4 P(t)^3} \\
&\quad - \frac{2Z\dot{P}(t) P(t) \left( a - e^{-Zt/\tau_{Ohm}} \right)^2 e^{2Zt/\tau_{Ohm}}}{\tau_{Ohm}} \cdot \frac{a e^{-2Zt/\tau_{Ohm}}}{\left( a - e^{-Zt/\tau_{Ohm}} \right)^3 P(t)} \\
&= -\dot{P}^2(t) P(t)^{-1} - \frac{2Za\dot{P}(t)}{\tau_{Ohm} \left( a - e^{-2t/\tau_{Ohm}} \right)} . \tag{26}
\end{aligned}
$$

Rearranging Equation (26), we obtain a very useful expression

$$\frac{\ddot{P}(t)}{\dot{P}(t)} + \frac{\dot{P}(t)}{P(t)} = \frac{-2Z \left( 1 + \frac{1}{\omega_B \tau} \right)}{\left( 1 + \frac{1}{\omega_B \tau} - \exp(-Zt/\tau_{Ohm}) \right) \tau_{Ohm}} . \tag{27}$$

Inserting Equation (9) into Equation (27), we obtain

$$(n - 3) \frac{\dot{P}(t)}{P} = \frac{-2Z \left( 1 + \frac{1}{\omega_B \tau} \right)}{\left( 1 + \frac{1}{\omega_B \tau} - \exp(-Zt/\tau_{Ohm}) \right) \tau_{Ohm}} . \tag{28}$$

Inserting $P = 206.4$ ms, $\dot{P} = 9.7728 \times 10^{-13}\,\mathrm{s\,s^{-1}}$ and $\ddot{P} = -5.27(13) \times 10^{-24}\,\mathrm{s\,s^{-2}}$ into Equation (27), the left side of Equation (27) is equal to $-7.09 \times 10^{-13}\,\mathrm{s^{-1}}$. Substituting the gravitational redshift $Z = 0.9$ into Equation (27), then we make a plot of Ohmic dissipation timescale $\tau_{Ohm}$ versus magnetization parameter $\omega_B \tau$ by method of numerical simulations, as shown in Figure 2.

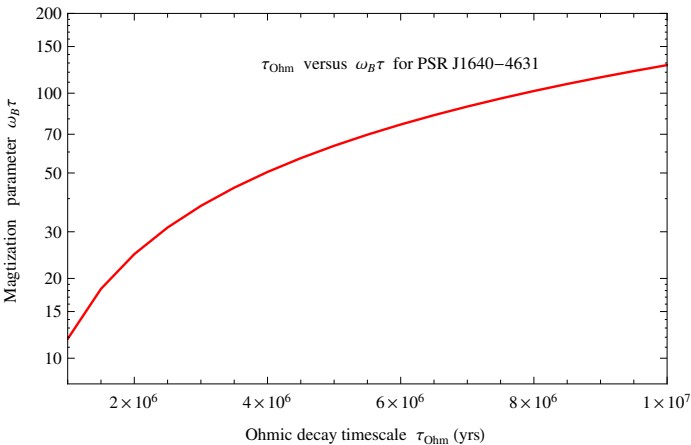

**Figure 2.** The relation of $\tau_{Ohm}$ and $\omega_B\tau$ for PSR J1640-4631.

Figure 2 clearly shows that $\omega_B\tau$ increases as $\tau_{Ohm}$ increases. Combining the magnetization parameter with Equation (28), the relation between the Ohmic timescale $\tau_{Ohm}$ and Hall drift timescale $\tau_{Hall}$ is obtained. Figure 3 shows that $\tau_{Hall}$ decreases with increasing $\tau_{Ohm}$.

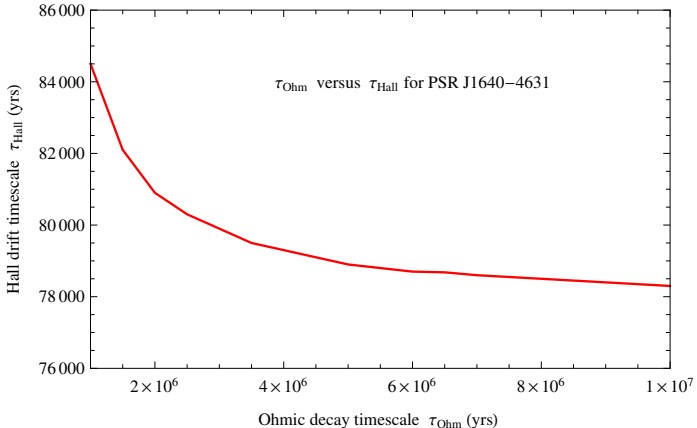

**Figure 3.** The relation of $\tau_{Ohm}$ and $\tau_{Hall}$ for PSR J1640-4631.

*3.3. Simulating the Dipole Magnetic Field Evolution and Rotation Evolution of PSR J1640-4631*

When $t = t_{age} = 3130$ yrs, combining the EoS and arrival time parameters, we obtain the present value of surface dipole magnetic field, $B_p(t_{age}) = (3c^3 I\dot{P}P/2\pi^2 R^6)^{1/2} \approx 2.305 \times 10^{13}$ G for PSR J1640-4631. Letting the Ohmic dissipation timescale vary between $1.0 \times 10^6$ yrs and $3.0 \times 10^7$ yrs. Inserting $P(t_{age}) = 0.2064$ ms and $B_p(t_{age}) = 2.305 \times 10^{13}$ G into the expressions of $B_p(t)$ and $P(t)$, we obtain the values of initial dipole magnetic field $B_0$ and initial spin period $P_0$. According to Equation (7), the Ohmic timescale is constrained as $\tau_{Ohm} \in (1.0 \times 10^6$–$9.4 \times 10^6)$ yrs. It is found that, when the Ohmic timescale is arbitrarily available in the range of $(1.0 \times 10^6$–$3.0 \times 10^7)$ yrs, the initial magnetic field $B_0$ ranges from $2.3752 \times 10^{13}$ G to $2.3810 \times 10^{13}$ G, in other words, $B_0$ is almost constant, and the initial period is distributed in a very narrow range, $P_0 \sim (37.8$–$40.6)$ ms. Then we obtain a mean dipole magnetic field decay rate of the pulsar $\Delta B_P/\Delta t = [B_P(t_{age}) - B_0]/\Delta t \approx -(2.3$–$2.4)\times 10^8$ G yr$^{-1}$. Some of simulation results are listed in Table 6.

**Table 6.** Partial fitted values of Ohmic dissipation timescale, Hall drift timescale, magnetization parameter, initial dipole magnetic field, and initial rotational period of PSR J 1640-4631.

| $\tau_{Ohm}$ (yrs) | $\tau_{Hall}$ (yrs) | $\omega_B \tau$ | $B_0$ (G) | $\omega_B \tau$ $(B_0/(10^{13}\,\text{G}))$ | $P_0$ (ms) |
|---|---|---|---|---|---|
| 1.0e6 | 8.45e4 | 11.83 | 2.3752e13 | 4.96 | 40.6 |
| 3.0e6 | 7.99e4 | 37.57 | 2.3758e13 | 15.8 | 40.4 |
| 5.0e6 | 7.89e4 | 63.31 | 2.3768e13 | 26.5 | 39.9 |
| 6.0e6 | 7.88e4 | 76.16 | 2.3769e13 | 31.9 | 39.7 |
| 7.0e6 | 7.86e4 | 89.05 | 2.3772e13 | 37.3 | 39.4 |
| 8.0e6 | 7.85e4 | 101.9 | 2.3774e13 | 42.7 | 39.2 |
| 9.0e6 | 7.84e4 | 114.8 | 2.3778e13 | 48.1 | 38.9 |
| 1.0e7 | 7.83e4 | 127.6 | 2.3782e13 | 53.5 | 38.4 |
| 2.0e7 | 7.80e4 | 256.3 | 2.3802e13 | 107.6 | 38.3 |
| 3.0e7 | 7.78e4 | 385.2 | 2.3810e13 | 161.2 | 37.8 |

Substituting Equations (19) and (23) simultaneously into Equation (28), we obtain the braking index expression

$$n = 3 + \frac{3Ic^3(\omega_B\tau)^2 Z}{\pi^2 R^6 B_0^2 \tau_{Ohm}} \cdot \frac{a(a - e^{-Zt/\tau_{Ohm}})}{e^{-2Zt/\tau_{Ohm}}} \left[ P_0^2 + \frac{4\pi^2 R^6 B_0^2 \tau_{Ohm}}{3Ic^3(\omega_B\tau)^2 Z} [\ln(a-1) \right.$$

$$\left. + \frac{a}{a-1} - \frac{a}{a - e^{-Zt/\tau_{Ohm}}} - \ln(a - e^{-Zt/\tau_{Ohm}})] \right] . \tag{29}$$

From Equation (29), we obtain the relation of the braking index $n$ and time $t$ for PSR J1640-4631, as shown in Figure 4.

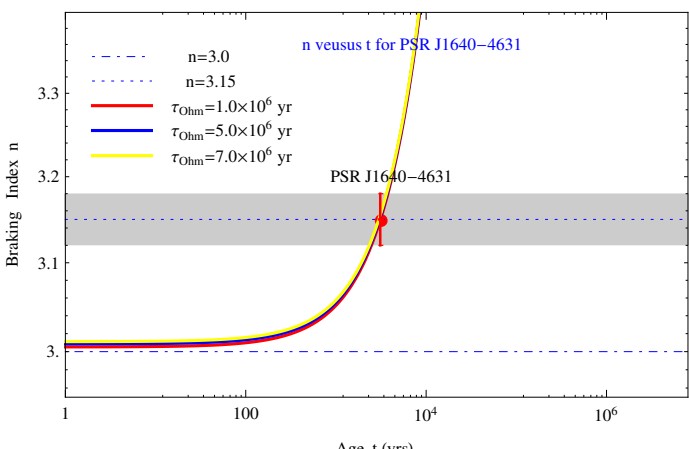

**Figure 4.** Braking index as a function of $t$ for PSR J1640-4631. Here, the measured value of $n$ is shown with the red dot.

In Figure 4, the blue dot-dashed line stands for the prediction of the MDR model, the horizontal blue dotted line and the surrounding shaded region denote, respectively, the measured braking index of $n = 3.15$ and its possible range given by the uncertainty 0.03 of the star. The solid red line represents the change trend expected by the dipole magnetic field decay model in the case of $\tau_{Ohm} = 1.0 \times 10^6$ yrs, $P_0 = 40.6$ ms and $B_0 = 2.3752 \times 10^{13}$ G, while the solid blue line represents thhe change trend expected by the dipole magnetic field decay model in the case of $\tau_{Ohm} = 5.0 \times 10^6$ yrs, $P_0 = 39.9$ ms and $B_0 = 2.3768 \times 10^{13}$ G, the solid yellow line represents the change trend expected by the dipole magnetic field decay model in the case of $\tau_{Ohm} = 7.0 \times 10^6$ yrs, $P_0 = 39.4$ ms and $B_0 = 2.3772 \times 10^{13}$ G. As can be seen from Figure 4, the braking index $n$ increases with the increase of $t$, due to the decay of the dipole magnetic field.

We are more concerned with the dipole magnetic field evolution of PSR J1640-4631. Here we select arbitrarily four different magnetization parameters (1) $\omega_B \tau = 5B_0/(10^{13}\,\mathrm{G})$, corresponding to $\tau_{Ohm} = 1.01 \times 10^6$ yrs, $\tau_{Hall} = 8.44 \times 10^4$ yrs and $B_0/(10^{13}\,\mathrm{G}) = 2.383$; (2) $\omega_B \tau = 20B_0/(10^{13}\,\mathrm{G})$, corresponding to $\tau_{Ohm} = 4.1 \times 10^6$ yrs, $\tau_{Hall} = 7.94 \times 10^4$ yrs and $B_0/(10^{13}\,\mathrm{G}) = 2.384$; (3) $\omega_B \tau = 30B_0/(10^{13}\,\mathrm{G})$ corresponding to $\tau_{Ohm} = 5.6 \times 10^6$ yrs, $\tau_{Hall} = 7.88 \times 10^4$ yrs and $B_0/(10^{13}\,\mathrm{G}) = 2.385$; (4) $\omega_B \tau = 45B_0/(10^{13}\,\mathrm{G})$ corresponding to $\tau_{Ohm} = 8.41 \times 10^6$ yrs, $\tau_{Hall} = 7.84 \times 10^4$ yrs and $B_0/(10^{13}\,\mathrm{G}) = 2.385$; From Equation (13), we plot the diagrams of $B_p$ versus $t$ for the pulsar in Figure 5.

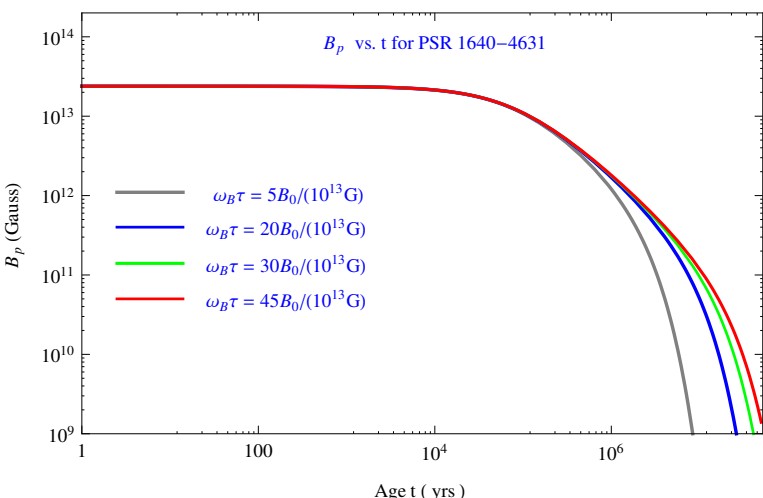

**Figure 5.** The dipolar magnetic field $B_p$ as a function of t for PSR J1640-4631 with $n > 3$.

From Equation (13), we plot the diagrams of $B_p$ versus $t$ for the pulsar in Figure 5. As can be seen from Figure 5, $B_p$ decreases with the increase in time $t$. The decay rate of the dipole magnetic field is an important parameter. According to Equation (14), we make a polt of $dB_P/dt$ and time $t$, as shown in Figure 6.

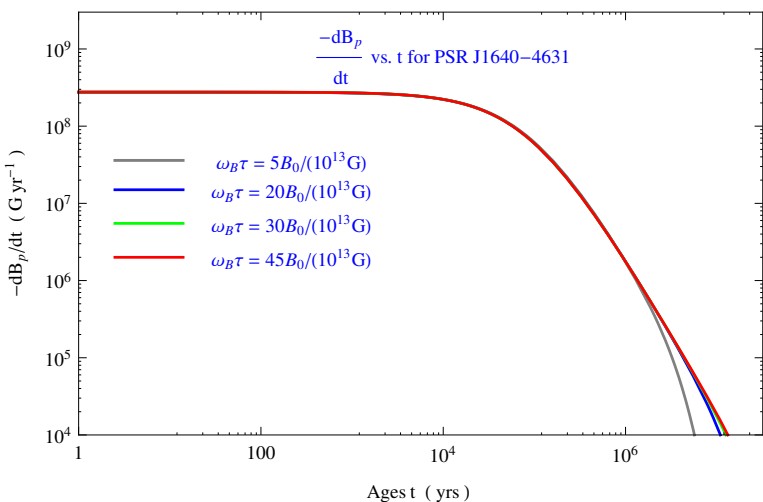

**Figure 6.** The dipolar magnetic field decay rate $dB_p/dt$ as a function of $t$ for PSR J1640-4631 with $n > 3$.

Figure 6 shows that $dB_P/dt$ decreases with the increase in $t$. With the combination of Ohmic dissipation and Hall drift, the dipole magnetic energy decay rate, $L_p$, is then estimated as

$$
\begin{aligned}
L_p &= \frac{-1}{4\pi} \int_V B_p \frac{dB_p}{dt} dV \\
&= \frac{1}{4\pi} \int_V \left( \frac{B_p^2}{\tau_{Ohm}} + \frac{B_p^3}{\tau_{Hall} B_0} \right) dV \\
&= \int_V Z B_0^2 \left[ \frac{e^{-2Zt/\tau_{Ohm}}}{[(1+\omega_B\tau(1-e^{-Zt/\tau_{Ohm}})]^2} + \frac{e^{-3Zt/\tau_{Ohm}}}{\tau_{Hall}} [(1+\omega_B\tau(1-e^{-Zt/\tau_{Ohm}})]^3 \right] dV \,, \quad (30)
\end{aligned}
$$

where $dV = 4\pi r^2 dr$, the thickness of the NS inner crust is $R_c \approx 0.7\,\mathrm{km}$, so the ratio of distance $r/R \sim (0.92\text{–}1.0)$ with $r$ the distance from the layer to the star's center. We numerically simulate the relation of the dipole magnetic field energy decay rate $L_p$ versus $t$, as shown in Figure 7.

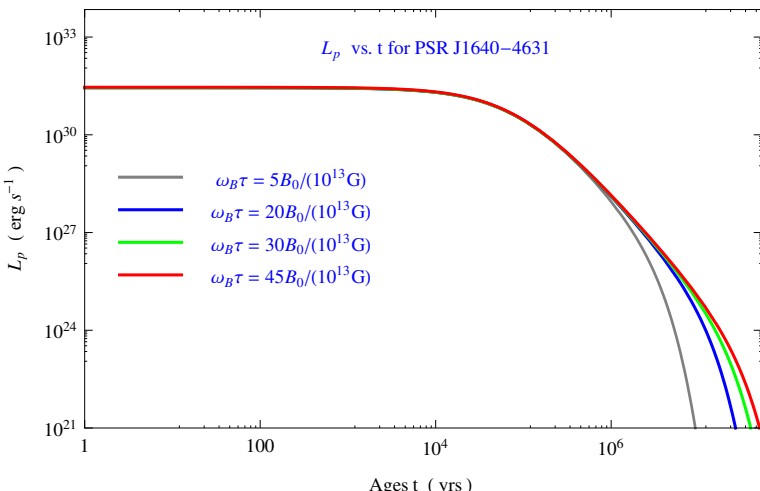

**Figure 7.** The dipolar magnetic field decay rate $L_p$ as a function of $t$ for PSR J1640-4631 with $n > 3$.

It is obvious that $L_p$ decreases with the increase in $t$. The differences among three fitted curves are very small, and each curve changes from smoother to steeper, as shown in Figures 5–7. In addition, we calculate the present values of $dB_P/dt$ and $L_p$: (1) when $\omega_B\tau = 5B_0/(10^{13}\,\mathrm{G})$, $dB_p/dt = -2.84 \times 10^8\,\mathrm{G\,yr^{-1}}$, and $L_p/dt = 2.67 \times 10^{31}\,\mathrm{erg\,s^{-1}}$; (2) when $\omega_B\tau = 20B_0/(10^{13}\,\mathrm{G})$, $dB_p/dt = -2.86 \times 10^8\,\mathrm{G\,yr^{-1}}$, and $L_p/dt = 2.71 \times 10^{31}\,\mathrm{erg\,s^{-1}}$; (3) when $\omega_B\tau = 30B_0/(10^{13}\,\mathrm{G})$, $dB_p/dt = -2.88 \times 10^8\,\mathrm{G\,yr^{-1}}$, and $L_p/dt = 2.78 \times 10^{31}\,\mathrm{erg\,s^{-1}}$; (4) when $\omega_B\tau = 45B_0/(10^{13}\,\mathrm{G})$, $dB_p/dt = -2.91 \times 10^8\,\mathrm{G\,yr^{-1}}$, and $L_p/dt = 2.80 \times 10^{31}\,\mathrm{erg\,s^{-1}}$. It is clear that the dipole magnetic field change rate and magnetic field energy release rate are almost constant (the relative increase rates of $\Delta \dot{B}_p / \dot{B}_p$ and $\Delta L_p / L_p$ are less than 5%).

## 4. Summary and Comparisons

In this work, we first introduce two different fiducial dipole magnetic fields: $B = 1.0 \times 10^{13}$ G and assume that the internal temperature of young highly magnetized NSs cools from $T = 10^9$ K to $1.0 \times 10^8$ K in the first few million years, then calculate the conductivity and magnetization parameter in the NS inner crust and give a general expression of magnetization parameter, magnetic field decay timescales, and magnetic field strength.

As the promotion and application of Equation (7), we assume that the high-braking-index pulsar PSR J1640-4631 is experiencing a dipole magnetic field decay, but the dipole braking still dominates, then establish a theoretical model, and give a constrained Ohmic decay timescale $\tau_{Ohm} \sim (1.0 \times 10^6\text{–}9.4 \times 10^6)$ yrs. At last, we numerically simulate the dipole magnetic field

evolution and spin-down evolution of PSR J1640-4631, and compare the fitting results with the observations of the star. Interestingly, in Reference [9], the authors also gave a very practical expression $\tau_{Ohm}/\tau_{Hall} = \omega_B \tau = (1$–$10)B_0/(10^{13}\,\text{G})$. However, Reference [9] ignored the relevant calculations when temperature drops to $T = 10^7\,\text{K}$, and adopted a relatively low impurity concentration range $Q \sim (10^{-4}$–$10^{-2})$. They also did not consider the effect of magnetic field on $Q$. Thus, the general expression of magnetization parameter given by this paper may be more close to the actual situation of NSs, compared with that given by Reference [9].

Very recently, in Reference [13], the authors introduced an effective dipole magnetic field decay time scale and adopted the neutron star mass $M \sim (1.0$–$2.2)M_\odot$ (corresponding to the moment of inertia $I \sim (0.8$–$2.1) \times 10^{45}\,\text{g}\,\text{cm}^2)$ in the EoS, and calculated the initial dipole magnetic field range $B_0 \sim (1.84$–$4.20) \times 10^{13}\,\text{G}$ for PSR J1640-4631, the initial rotation period range $P_0 \sim (17$–$44)\,\text{ms}$, and the magnetic field decay rate range $dB_p/dt = -(1.16$–$3.85) \times 10^8\,\text{G}\,\text{yr}^{-1}$. In order to account for the high braking index of PSR J1640-4631 with a combination of the magneto dipole radiation and dipole magnetic field decay models, Reference [13] introduced a mean rotation energy conversion coefficient $\bar{\varsigma}$, and adopted the APR3 model, which provides a constraint on the maximum NS mass $M_{max} \leq 2.2M_\odot$. By introducing an effective dipole magnetic field decay timescale $\tau_D$, They selected the NS mass $M \sim (1$–$2.2)\,M_\odot$, corresponding to $I \sim (0.8$–$2.09) \times 10^{45}\,\text{g}\,\text{cm}^2$, then calculated the star's initial dipole field, $B_0 \sim (1.84$–$4.20) \times 10^{13}\,\text{G}$, the initial spin period $P_0 \sim 17$–$44$ ms, and the magnetic field decay rate $dB_P/dt = -(1.16$–$3.85) \times 10^8\,\text{G}\,\text{yr}^{-1}$. However, the authors only adopted a simple exponential magnetic field evolution model, and introduced an effective dipole magnetic field decay timescale $\tau_D$, which replaces the special calculations of Ohmic decay timescale and Hall drift timescale, and did not use magnetization parameter to effectively limit the two timescales. Although the calculations in Reference [13] are basically consistent with our results, our results may be more reliable than those in Reference [13]. This study is expected to apply to more young pulsars and will be tested in the future observations.

**Author Contributions:** H.W. is responsible for organizing, writing and modifying article; Z.-F.G. is responsible for arranging article layout structure, choosing references and writing cover letter; H.-Y.J. is responsible for theoretical analysis, and data processing; N.W. is responsible for making plots and tables and improving language, and X.-D.L. is responsible for theoretical analysis and publication fee. All authors have read and agreed to the published version of the manuscript.

**Funding:** This work was supported by the National Key Research and Development Program of China (2018YFA04040602, 2018YFA0404202, and 2016YFA0400803), the Natural Science Foundation of China under grant No.11673056, 11773015, 11173042 and 11947404, Xinjiang Natural ScienceFoundation No.2018D01A24 and Project U1838201 supported by NSFC and CAS.

**Acknowledgments:** We are grateful the referees for helpful comments.

**Conflicts of Interest:** The authors declare that there are no conflict of interest.

## Appendix A

**Table A1.** Table 1 Partial values of electrical conductivity and magnetization parameter for two different temperatures T and two different impurity parameters $Q$ in the inner crust of NSs for the nuclear mass models HFB-22, HFB-24 and HFB-26. The unit of magnetization parameter $\omega_B \tau$ is the normalized magnetic field $B_{14}$ when the dipolar magnetic field strength $B = 1.0 \times 10^{14}$ G. Here the crust is assumed to be isothermal.

| | | | | 5e8 K | | | | 1.0e9 K | | | |
|---|---|---|---|---|---|---|---|---|---|---|---|
| | | | | $Q = 0.01$ | | $Q = 1$ | | $Q = 0.01$ | | $Q = 1$ | |
| $\bar{n}_b$ (fm$^{-3}$) | $\rho$ (g cm$^{-3}$) | $Y_e$ | $n_e$ (cm$^{-3}$) | $\sigma$ (s$^{-1}$) | $\omega_B \tau$ | $\sigma$ (s$^{-1}$) | $\omega_B \tau$ | $\sigma$ (s$^{-1}$) | $\omega_B \tau$ | $\sigma$ (s$^{-1}$) | $\omega_B \tau$ |
| | | | | HFB-22 | | | | | | | |
| 2.700e-04 | 4.513e11 | 2.955e-01 | 7.979e34 | 6.54e22 | 5.73 | 6.53e22 | 5.69 | 3.54e22 | 3.08 | 3.52e22 | 3.07 |
| 5.253e-04 | 8.790e11 | 1.839e-01 | 9.658e34 | 7.68e22 | 5.53 | 7.62e22 | 5.48 | 4.07e22 | 2.93 | 4.05e22 | 2.92 |
| 8.778e-04 | 1.470e12 | 1.294e-01 | 1.136e35 | 8.83e22 | 5.41 | 8.77e22 | 5.37 | 4.62e22 | 2.83 | 4.60e22 | 2.81 |
| 1.593e-03 | 2.670e12 | 8.832e-02 | 1.407e35 | 1.09e23 | 5.35 | 1.07e23 | 5.24 | 5.53e22 | 2.73 | 5.49e22 | 2.71 |
| 2.094e-03 | 3.510e12 | 7.511e-02 | 1.573e35 | 1.21e23 | 5.33 | 1.20e23 | 5.26 | 6.08e22 | 2.69 | 6.05e22 | 2.68 |
| 2.707e-03 | 4.540e12 | 6.510e-02 | 1.762e35 | 1.35e23 | 5.29 | 1.34e23 | 5.25 | 6.74e22 | 2.66 | 6.70e22 | 2.65 |
| 3.726e-03 | 6.250e12 | 5.526e-02 | 2.059e35 | 1.59e23 | 5.34 | 1.57e23 | 5.27 | 7.81e22 | 2.64 | 7.76e22 | 2.62 |
| 4.994e-03 | 8.380e12 | 4.826e-02 | 2.410e35 | 1.88e23 | 5.42 | 1.85e23 | 5.33 | 9.11e22 | 2.62 | 9.05e22 | 2.60 |
| 8.931e-03 | 1.500e13 | 3.841e-02 | 3.431e35 | 2.86e23 | 5.79 | 2.81e23 | 5.69 | 1.33e23 | 2.69 | 1.32e23 | 2.67 |
| 1.535e-02 | 2.580e13 | 3.221e-02 | 4.944e35 | 4.75e23 | 6.67 | 4.63e23 | 6.50 | 2.11e23 | 2.96 | 2.08e23 | 2.92 |
| 2.781e-02 | 4.680e13 | 2.696e-02 | 7.498e35 | 9.85e23 | 9.12 | 9.41e23 | 8.72 | 4.10e23 | 3.80 | 4.02e23 | 3.72 |
| 3.400e-02 | 5.725e13 | 2.562e-02 | 8.709e35 | 1.36e24 | 10.84 | 1.28e24 | 10.21 | 5.51e23 | 4.39 | 5.38e23 | 4.29 |
| | | | | HFB-24 | | | | | | | |
| 2.570e-04 | 4.296e11 | 3.028e-01 | 7.783e34 | 6.46e22 | 5.77 | 6.41e22 | 5.72 | 3.48e22 | 3.11 | 3.46e22 | 3.09 |
| 2.788e-04 | 4.660e11 | 2.859e-01 | 7.970e34 | 6.58e22 | 5.74 | 6.53e22 | 5.69 | 3.54e22 | 3.09 | 3.52e22 | 3.07 |
| 5.253e-04 | 8.790e11 | 1.847e-01 | 9.703e34 | 7.72e22 | 5.53 | 7.65e22 | 5.48 | 4.08e22 | 2.93 | 4.07e22 | 2.91 |
| 8.778e-04 | 1.470e12 | 1.325e-01 | 1.163e35 | 9.02e22 | 5.39 | 8.95e22 | 5.34 | 4.70e22 | 2.82 | 4.67e22 | 2.79 |
| 1.194e-03 | 2.000e12 | 1.100e-01 | 1.314e35 | 1.02e23 | 5.35 | 9.98e22 | 5.28 | 5.19e22 | 2.75 | 5.16e22 | 2.72 |
| 1.593e-03 | 2.670e12 | 9.352e-02 | 1.490e35 | 1.14e23 | 5.28 | 1.13e23 | 5.23 | 5.77e22 | 2.69 | 5.74e22 | 2.68 |
| 2.707e-03 | 4.540e12 | 7.186e-02 | 1.945e35 | 1.49e23 | 5.29 | 1.47e23 | 5.22 | 7.34e22 | 2.63 | 7.30e22 | 2.61 |
| 3.724e-03 | 6.250e12 | 6.288e-02 | 2.342e35 | 1.80e23 | 5.34 | 1.78e23 | 5.28 | 8.77e22 | 2.60 | 8.71e22 | 2.59 |
| 4.991e-03 | 8.380e12 | 5.661e-02 | 2.825e35 | 2.23e23 | 5.48 | 2.20e23 | 5.41 | 1.06e23 | 2.61 | 1.05e23 | 2.58 |
| 8.926e-03 | 1.500e13 | 4.809e-02 | 4.293e35 | 3.77e23 | 6.10 | 3.69e23 | 5.97 | 1.71e23 | 2.77 | 1.69e23 | 2.73 |
| 1.534e-02 | 2.580e13 | 4.289e-02 | 6.578e35 | 7.07e23 | 7.46 | 6.83e23 | 7.21 | 3.03e23 | 3.20 | 2.98e23 | 3.15 |
| 2.778e-02 | 4.680e13 | 3.795e-02 | 1.054e36 | 1.66e24 | 10.94 | 1.55e24 | 10.21 | 6.64e23 | 4.37 | 6.46e23 | 4.26 |
| 3.000e-02 | 5.055e13 | 3.729e-02 | 1.119e36 | 1.87e24 | 11.61 | 1.74e24 | 10.80 | 7.44e23 | 4.62 | 7.23e23 | 4.49 |
| 5.000e-02 | 8.437e13 | 3.328e-02 | 1.664e36 | 4.93e24 | 20.57 | 4.27e24 | 17.82 | 1.83e24 | 7.64 | 1.73e24 | 7.22 |
| 5.634e-02 * | 9.510e13 * | 3.279e-02 | 1.847e36 | 6.65e24 | 25.00 | 5.57e24 | 20.94 | 2.41e24 | 9.06 | 2.25e24 | 8.46 |
| | | | | HFB-26 | | | | | | | |
| 2.620e-04 | 4.379e11 | 2.996e-01 | 7.850e34 | 6.49e22 | 5.74 | 6.44e22 | 5.70 | 3.48e22 | 3.09 | 3.48e22 | 3.08 |
| 2.788e-04 | 4.660e11 | 2.866e-01 | 7.988e34 | 6.58e22 | 5.72 | 6.53e22 | 5.68 | 3.54e22 | 3.08 | 3.52e22 | 3.06 |
| 5.252e-04 | 8.790e11 | 1.833e-01 | 9.629e34 | 7.66e22 | 5.53 | 7.59e22 | 5.48 | 4.04e22 | 2.93 | 4.04e22 | 2.92 |
| 8.777e-04 | 1.470e12 | 1.298e-01 | 1.139e35 | 8.84e22 | 5.39 | 8.77e22 | 5.35 | 4.61e22 | 2.81 | 4.60e22 | 2.80 |
| 1.194e-03 | 2.000e12 | 1.066e-01 | 1.273e35 | 9.78e22 | 5.33 | 9.68e22 | 5.29 | 5.06e22 | 2.76 | 5.02e22 | 2.74 |
| 1.593e-03 | 2.670e12 | 8.955e-02 | 1.427e35 | 1.10e23 | 5.31 | 1.08e23 | 5.22 | 5.56e22 | 2.71 | 5.52e22 | 2.69 |
| 2.707e-03 | 4.540e12 | 6.698e-02 | 1.813e35 | 1.38e23 | 5.26 | 1.37e23 | 5.22 | 6.87e22 | 2.64 | 6.83e22 | 2.62 |
| 3.726e-03 | 6.250e12 | 5.756e-02 | 2.144e35 | 1.64e23 | 5.31 | 1.61e23 | 5.21 | 8.04e22 | 2.61 | 7.98e22 | 2.58 |
| 4.993e-03 | 8.380e12 | 5.098e-02 | 2.546e35 | 1.97e23 | 5.37 | 1.95e23 | 5.32 | 9.51e22 | 2.59 | 9.44e22 | 2.57 |
| 8.929e-03 | 1.500e13 | 4.224e-02 | 3.772e35 | 3.16e23 | 5.82 | 3.10e23 | 5.71 | 1.45e23 | 2.67 | 1.44e23 | 2.65 |
| 1.534e-02 | 2.580e13 | 3.758e-02 | 5.765e35 | 5.70e23 | 6.87 | 5.53e23 | 6.66 | 2.47e23 | 2.98 | 2.44e23 | 2.94 |
| 2.778e-02 | 4.680e13 | 3.456e-02 | 9.601e35 | 1.36e24 | 9.84 | 1.28e24 | 9.26 | 5.41e23 | 3.91 | 5.29e23 | 3.83 |
| 3.000e-02 | 5.054e13 | 3.425e-02 | 1.028e36 | 1.55e24 | 10.47 | 1.45e24 | 9.80 | 6.11e23 | 4.13 | 5.96e23 | 4.03 |
| 5.000e-02 | 8.437e13 | 3.278e-02 | 1.639e36 | 4.44e24 | 18.81 | 3.89e24 | 16.48 | 1.62e24 | 6.86 | 1.54e24 | 6.52 |
| 5.634e-02 * | 9.510e13 * | 3.274e-02 | 1.845e36 | 6.08e24 | 22.88 | 5.15e24 | 19.38 | 2.16e24 | 8.13 | 2.03e24 | 7.64 |

\* The sign denotes that the computed equilibrium proton number $Z_{eq}$ begins to deviate from a standard value of $Z = 40$.

**Table A2.** Table 1 Partial values of electrical conductivity and magnetization parameter for two different temperatures T and two different impurity parameters $Q$ in the inner crust of NSs for the nuclear mass models HFB-22, HFB-24 and HFB-26. The unit of magnetization parameter $\omega_B\tau$ is the normalized magnetic field $B_{14}$ when the dipolar magnetic field strength $B = 1.0 \times 10^{14}$ G. Here the crust is assumed to be isothermal.

| | | | | 1e7 K | | | | 1.0e8 K | | | |
|---|---|---|---|---|---|---|---|---|---|---|---|
| | | | | $Q = 0.01$ | | $Q = 1$ | | $Q = 0.01$ | | $Q = 1$ | |
| $\bar{n}_b$ (fm$^{-3}$) | $\rho$ (g cm$^{-3}$) | $Y_e$ | $n_e$ (cm$^{-3}$) | $\sigma$ (s$^{-1}$) | $\omega_B\tau$ | $\sigma$ (s$^{-1}$) | $\omega_B\tau$ | $\sigma$ (s$^{-1}$) | $\omega_B\tau$ | $\sigma$ (s$^{-1}$) | $\omega_B\tau$ |
| | | | | HFB-22 | | | | | | | |
| 2.700e-04 | 4.513e11 | 2.955e-01 | 7.979e34 | 4.78e25 | 4162.2 | 7.40e24 | 647.2 | 5.66e23 | 49.3 | 5.32e23 | 46.3 |
| 5.253e-04 | 8.790e11 | 1.839e-01 | 9.658e34 | 6.20e25 | 4461.2 | 8.14e24 | 588.5 | 7.02e23 | 50.5 | 6.52e23 | 46.9 |
| 8.778e-04 | 1.470e12 | 1.294e-01 | 1.136e35 | 7.66e25 | 4688.2 | 8.79e24 | 541.2 | 8.53e23 | 52.2 | 7.86e23 | 48.0 |
| 1.593e-03 | 2.670e12 | 8.832e-02 | 1.407e35 | 1.03e26 | 5052.5 | 9.70e24 | 480.2 | 1.12e24 | 54.9 | 1.02e24 | 49.9 |
| 2.094e-03 | 3.510e12 | 7.511e-02 | 1.573e35 | 1.19e26 | 5222.9 | 1.03e25 | 452.3 | 1.29e24 | 56.7 | 1.16e24 | 51.0 |
| 2.707e-03 | 4.540e12 | 6.510e-02 | 1.762e35 | 1.38e26 | 5441.2 | 1.07e25 | 421.7 | 1.50e24 | 58.9 | 1.33e24 | 52.2 |
| 3.726e-03 | 6.250e12 | 5.526e-02 | 2.059e35 | 1.72e26 | 5810.2 | 1.15e25 | 387.9 | 1.83e24 | 61.7 | 1.59e24 | 53.7 |
| 4.994e-03 | 8.380e12 | 4.826e-02 | 2.410e35 | 2.16e26 | 6224.1 | 1.23e25 | 354.4 | 2.28e24 | 65.7 | 1.94e24 | 55.9 |
| 8.931e-03 | 1.500e13 | 3.841e-02 | 3.431e35 | 3.74e26 | 7569.9 | 1.44e25 | 291.5 | 3.89e24 | 78.7 | 3.09e24 | 62.5 |
| 1.535e-02 | 2.580e13 | 3.221e-02 | 4.944e35 | 7.06e26 | 9916.6 | 1.70e25 | 238.7 | 7.24e24 | 101.7 | 5.11e24 | 71.8 |
| 2.781e-02 | 4.680e13 | 2.696e-02 | 7.498e35 | 1.66e27 | 15,374 | 2.08e25 | 192.6 | 1.69e25 | 156.5 | 9.36e24 | 86.7 |
| 3.400e-02 | 5.725e13 | 2.562e-02 | 8.709e35 | 2.39e27 | 19,058 | 2.25e25 | 179.4 | 2.43e25 | 193.8 | 1.17e25 | 93.3 |
| | | | | HFB-24 | | | | | | | |
| 2.570e-04 | 4.296e11 | 3.028e-01 | 7.783e34 | 4.70e25 | 4182.2 | 7.31324 | 654.5 | 5.55e23 | 49.5 | 5.22e23 | 46.6 |
| 2.788e-04 | 4.660e11 | 2.859e-01 | 7.970e34 | 4.83e25 | 4198.3 | 7.41e24 | 645.7 | 5.69e23 | 49.6 | 5.35e23 | 46.6 |
| 5.253e-04 | 8.790e11 | 1.847e-01 | 9.703e34 | 6.30e25 | 4498.1 | 8.16e24 | 585.8 | 7.10e23 | 50.9 | 6.60e23 | 47.3 |
| 8.778e-04 | 1.470e12 | 1.325e-01 | 1.163e35 | 7.98e25 | 4762.4 | 8.89e24 | 533.6 | 8.84e23 | 52.9 | 8.11e23 | 48.6 |
| 1.194e-03 | 2.000e12 | 1.100e-01 | 1.314e35 | 9.38e25 | 4956.5 | 9.40e24 | 499.5 | 1.05e24 | 54.6 | 9.36e23 | 49.5 |
| 1.593e-03 | 2.670e12 | 9.352e-02 | 1.490e35 | 1.12e26 | 5178.6 | 9.95e24 | 465.2 | 1.21e24 | 56.1 | 1.10e24 | 50.9 |
| 2.707e-03 | 4.540e12 | 7.186e-02 | 1.945e35 | 1.61e26 | 5719.2 | 1.13e25 | 402.2 | 1.72e24 | 61.2 | 1.51e24 | 53.7 |
| 3.724e-03 | 6.250e12 | 6.288e-02 | 2.342e35 | 2.10e26 | 6197.2 | 1.23e25 | 361.8 | 2.21e24 | 65.2 | 1.89e24 | 55.7 |
| 4.991e-03 | 8.380e12 | 5.661e-02 | 2.825e35 | 2.77e26 | 6784.7 | 1.32e25 | 324.5 | 2.88e24 | 70.8 | 2.39e24 | 58.8 |
| 8.926e-03 | 1.500e13 | 4.809e-02 | 4.293e35 | 5.32e26 | 8621.9 | 1.59e25 | 257.2 | 5.49e24 | 88.8 | 4.11e24 | 66.5 |
| 1.534e-02 | 2.580e13 | 4.289e-02 | 6.578e35 | 1.13e27 | 11,929 | 1.92e25 | 202.7 | 1.15e25 | 121.4 | 7.25e24 | 76.5 |
| 2.778e-02 | 4.680e13 | 3.795e-02 | 1.054e36 | 2.96e27 | 19,502 | 2.42e25 | 159.4 | 3.00e25 | 197.7 | 1.35e25 | 88.9 |
| 3.000e-02 | 5.055e13 | 3.729e-02 | 1.119e36 | 3.38e27 | 20,976 | 2.50e25 | 155.1 | 3.44e25 | 213.5 | 1.45e25 | 90.0 |
| 5.000e-02 | 8.437e13 | 3.328e-02 | 1.664e36 | 9.59e27 | 40,022 | 3.19e25 | 133.1 | 9.85e25 | 411.1 | 2.41e25 | 101.0 |
| 5.634e-02 * | 9.510e13 * | 3.279e-02 | 1.847e36 | 1.32e28 | 49,630 | 3.43e25 | 129.0 | 1.36e26 | 511.3 | 2.74e25 | 103.0 |
| | | | | HFB-26 | | | | | | | |
| 2.620e-04 | 4.379e11 | 2.996e-01 | 7.850e34 | 4.72e25 | 4162.2 | 7.35e24 | 649.6 | 5.57e23 | 49.2 | 5.24e23 | 46.4 |
| 2.788e-04 | 4.660e11 | 2.866e-01 | 7.988e34 | 4.82e25 | 4175.3 | 7.41e24 | 646.9 | 5.68e23 | 49.3 | 5.34e23 | 46.4 |
| 5.252e-04 | 8.790e11 | 1.833e-01 | 9.629e34 | 6.24e25 | 4488.8 | 8.13e24 | 587.2 | 7.03e23 | 50.7 | 6.54e23 | 47.2 |
| 8.777e-04 | 1.470e12 | 1.298e-01 | 1.139e35 | 7.75e25 | 4721.3 | 8.79e24 | 537.5 | 8.60e23 | 52.5 | 7.91e23 | 48.3 |
| 1.194e-03 | 2.000e12 | 1.066e-01 | 1.273e35 | 8.99e25 | 4895.8 | 9.26e24 | 506.1 | 9.87e23 | 53.8 | 9.01e23 | 49.1 |
| 1.593e-03 | 2.670e12 | 8.955e-02 | 1.427e35 | 1.06e26 | 5112.0 | 9.72e24 | 475.5 | 1.15e24 | 55.6 | 1.04e24 | 50.2 |
| 2.707e-03 | 4.540e12 | 6.698e-02 | 1.813e35 | 1.46e26 | 5556.1 | 1.10e25 | 419.6 | 1.56e24 | 59.5 | 1.38e24 | 52.6 |
| 3.726e-03 | 6.250e12 | 5.756e-02 | 2.144e35 | 1.84e26 | 5959.8 | 1.17e25 | 379.0 | 1.95e24 | 63.2 | 1.69e24 | 54.4 |
| 4.993e-03 | 8.380e12 | 5.098e-02 | 2.546e35 | 2.36e26 | 6437.1 | 1.26e25 | 343.7 | 2.48e24 | 67.6 | 2.09e24 | 57.0 |
| 8.929e-03 | 1.500e13 | 4.224e-02 | 3.772e35 | 4.33e26 | 7971.8 | 1.50e25 | 276.2 | 4.47e24 | 82.3 | 3.47e24 | 63.9 |
| 1.534e-02 | 2.580e13 | 3.758e-02 | 5.765e35 | 8.95e26 | 10,781 | 1.80e25 | 216.8 | 9.13e24 | 110.0 | 6.10e24 | 73.5 |
| 2.778e-02 | 4.680e13 | 3.456e-02 | 9.601e35 | 2.44e27 | 17,649 | 2.29e25 | 165.6 | 2.48e25 | 179.4 | 1.20e25 | 86.8 |
| 3.000e-02 | 5.054e13 | 3.425e-02 | 1.028e36 | 2.83e27 | 19,117 | 2.37e25 | 160.1 | 2.87e25 | 193.9 | 1.31e25 | 88.5 |
| 5.000e-02 | 8.437e13 | 3.278e-02 | 1.639e36 | 8.88e27 | 37,624 | 3.09e25 | 130.9 | 9.10e25 | 385.6 | 2.31e25 | 97.9 |
| 5.634e-02 * | 9.510e13 * | 3.274e-02 | 1.845e36 | 1.23e28 | 46,296 | 3.34e25 | 125.7 | 1.27e26 | 478.0 | 2.65e25 | 99.7 |

\* The sign denotes that the computed equilibrium proton number $Z_{eq}$ begins to deviate from a standard value of $Z = 40$.

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
