# Peer review of "Estimation of Electrical Conductivity and Magnetization Parameter of Neutron Star Crusts and Applied to the High-Braking-Index Pulsar PSR J1640-4631"

_universe, doi:10.3390/universe6050063_

Round 1
Reviewer 1 Report
The authors calculate the electrical conductivity in the inner crust of
a neutron star using a free computer code, convert it
into the effective relaxation time \tau and the Hall parameter
\omega_B\tau by Eq.(2), and use these values in a simple analytical
toy-model of the magnetic field evolution.
This approach is very simplistic. There are detailed numerical
simulations of the magnetic field evolution in the literature, recently
reviewed by Pons & Vigano (2019, Living Reviews in Comput. Astrophys.,
5, 3). The present authors do not mention these advances in the
Introduction (although they are aware of some of them, as one can see
from the list of references).
To calculate the conductivity, the authors use a computer code, which
was previously developed in a series of papers as reviewed by Potekhin,
Pons, & Page (2015, Space Sci. Rev., 191, 239). This is the basic
reference, as stated at the URL http://www.ioffe.ru/astro/conduct. The
authors give the correct URL, but wrong literature reference (line 111
of the manuscript). The authors state that the use of the modern
conductivity makes their results "more reliable ... compared with
previous calculations" in lines 113-114 and repeat almost the same
statement again in lines 115-116, but do not specify, which exactly
"previous calculations" are meant to be improved.
Moreover, in spite of the use of the modern computer code for the
conductivity, they employ outdated Baym-Bethe-Pethick (BBP) equation of
state. They reproduce a criticism of this equation of state from the
monograph of Shapiro & Teukolsky, who in turn refer to Canuto (1974).
The criticism is formulated by the present authors in an obscure manner,
although in the original publications it was clear: the realistic
ground-state charge number Z was argued to stay around 40 throughout the
crust, at sharp contrast to the BBP predictions. It is worth noticing
that some modern equations of state indeed show that Z is equal to 40
throughout the most part of the inner crust - e.g., Pearson et al.
(2018, MNRAS, 481, 2994). Therefore it would be more consistent to use
one of these modern equations of state, instead of just the verbal
criticism toward the classical pioneering BBP work.
In a strongly magnetized matter, electrical conductivity is a tensor.
The authors consider a scalar conductivity, implying actually the
longitudinal component (i.e., the largest eigenvalue) of this tensor.
The authors write in the abstract that the conductivity weakly depends
on a magnetic field. In fact, this dependence of the longitudinal
conductivity (which is caused by the quantum effects) usually is indeed
weak in the inner crust, but only until the field strength much exceeds
10^{15} G. Therefore the weak dependence is not absolute: it applies
only to relatively large densities and not too strong magnetic fields.
The numerical data in the tables and in the figures do not match each
other. For example, for magnetic field B=1.e13 G, at the lowest density
\rho=4.66e11 g/cm^3 and mild temperature T=5.e8 K, panels c and d of
Figure 1 show \omega\tau around 0.6 and 2.0, respectively, whereas
Tables 1 and A1 give \omega\tau=0.49 and 4.9 for the same B, \rho and T.
The Hall parameter \omega\tau is given in units of B_{13}, which is
awry, because \omega\tau is a dimensionless quantity.
Moreover, the numerical values in the tables and in the figures are
inaccurate. The inaccuracy increases with increasing density. For
example, let us consider Table 1, B=10^{13} G, Q=0.01, T=5.e8, the last
row \rho=1.3e14 g/cm^3. The authors correctly quote Z=120 and A=990 from
the BBP work in the second and third columns. According to BBP (their
Table 1), the density of nuclei is n_N = 1.78e-5 fm^{-3} = 1.78e34
cm^{-3}, which gives the number density of electrons n_e=n_N*Z=2.13e36
cm^{-3}, while the number in the third column of Table 1 is n_e=1.86e36
(besides, there is a typo in units at the top of this column). According
to BBP (their Tables 1 and 2), the total number of baryons per one
nucleus (i.e., per one Wigner-Seitz cell) at this density is
n_b/n_N=7.89e-2/1.78e-5=4433. Now the code condegin19.f at the
above-mentioned URL gives the longitudinal conductivity \sigma=3.04e24,
while the fourth column of Table 1 shows 9.21e23. Accordingly,
\omega\tau=0.995, while the fifth column of Table 1 shows 0.349. The
discrepancies in \sigma and \omega\tau can be explained by the use of
another code from the same Internet site, either conduct19.f or
condegen19.f, which have been designed exclusively for the outer crust.
These codes take neither the presence of free neutrons nor the nuclear
form factor (the finite sizes of nuclei) into account, and thus they may
be inaccurate in the inner crust of a neutron star.
Besides, there are mistypes and awkward English instances in the text.
Thus major improvements have to be made to the paper, before it may be
reconsidered for publication.
Reviewer 2 Report
In this paper the authors make estimations of the electrical conductivity and magnetization parameter of neutron star crusts. The authors use public programs to obtain conductivities and well known equation of state to build neutron star models. Different impurity parameters, temperatures and magnetic field strengths are used to obtain conductivity and magnetization parameter in the neutron star crust. A phenomenological simple model is used to describe the magnetic evolution of the star together with the rotational evolution. The model is applied to the pulsar PSR J1640-4631.
The paper might be useful for other researchers in the field and I am willing to recommend the paper for publication in Universe if the following points are clarify.
General comments
Throughout the paper the symbol $B_p$ is used sometimes to denote the magnetic field strength at the pole of the star (1) but some other times to denote the dipole field (eq. 3). Consistency is required.
Hall term leads to creation of toroidal field from poloidal one. However, the paper is restricted to poloidal field configuration. Some words on this point are in order.
From my point of view, \tau_Ohm and \tau_Hall are parameters to be obtained by using the phenomenological model (eqs. 1 and 3) and the data. Of course, these parameters depend on the microphysical phenomena, but they are a kind of average quantities in time and space. I do not see the needed to introduce the magnetization parameter in the game. The authors should explain this point.
Minor points.
pag. 3, eq. (3): The equation is wrong. Please, correct.
pag. 8, line 149: \tau_c instead of \tau.
pag. 8, line 146: braking.
pag. 10. eq. 13: B_p instead of B_\rho.
pag. 10, the line before eq. (14): instead of (1+t/\tau_{Hall}) should be (1-t/\tau_{Hall}).
pag. 10, eq(15): the symbol for the frequency should be the same on the left- and right-hand side. Should be B_p instead of B_s.
pag. 10, line 203: k instead of K.
pag. 11, eq. 28: The authors state that this equation is very useful. It would be interesting to write the equation introducing the braking index, and discuss the applicability of the equation in terms of this index.
pag. 12, fig.2 and 3: The figure captions are interchanged.
pag. 14, eq. 30. A dV is missing. What is the integration volume? It seems the integral is restricted to the crust of the star (line 253), but the magnetic field extends further. Explain.
Round 2
Reviewer 1 Report
The authors have carefully considered all comments and remarks and revised the paper accordingly. I believe the manuscript has been significantly improved and now warrants publication in Universe.
Reviewer 2 Report
The authors have answered the required questions and I recommend the paper for publication in Universe.